# Strain to shine: stretching-induced three-dimensional symmetries in nanoparticle-assembled photonic crystals

Tong An[1], Xinyu Jiang[1], Feng Gao[1], Christian Schäfer[2], Junjun Qiu[1], Nan Shi[1], Xiaokun Song[1], Manyao Zhang[1], Chris E. Finlayson[3], Xuezhi Zheng[4], Xiuhong Li[5], Feng Tian[5], Bin Zhu[6], Tan Sui[6], Xianhong Han[7], Jeremy J. Baumberg[8] ✉, Tongxiang Fan[1] ✉ & Qibin Zhao[1] ✉

Stretching elastic materials containing nanoparticle lattices is common in research and industrial settings, yet our knowledge of the deformation process remains limited. Understanding how such lattices reconfigure is critically important, as changes in microstructure lead to significant alterations in their performance. This understanding has been extremely difficult to achieve due to a lack of fundamental rules governing the rearrangements. Our study elucidates the physical processes and underlying mechanisms of three-dimensional lattice transformations in a polymeric photonic crystal from 0% to over 200% strain during uniaxial stretching. Corroborated by comprehensive experimental characterizations, we present analytical models that precisely predict both the three-dimensional lattice structures and the macroscale deformations throughout the stretching process. These models reveal how the nanoparticle lattice and matrix polymer jointly determine the resultant structures, which breaks the original structural symmetry and profoundly changes the dispersion of photonic bandgaps. Stretching induces shifting of the main pseudogap structure out from the 1st Brillouin zone and the merging of different symmetry points. Evolutions of multiple photonic bandgaps reveal potential optical singularities shifting with strain. This work sets a new benchmark for the reconfiguration of soft material structures and may lay the groundwork for the study of stretchable three-dimensional topological photonic crystals.

Nanoparticles are often assembled into three-dimensional (3D) lattices which provide unique properties from wavelength-scale periodic structures[1-4]. Due to the short-range interactions between the particles[5], materials comprised of pure particle lattices tend to be brittle and susceptible to damage under deformation without special treatment[6]. A notable approach to increase their robustness involves surrounding the nanoparticles with elastic polymers[7]. This enables the resultant materials to be strain-resilient and display tuneable properties upon strain[8], which finds wide applications in diverse fields[9]. These tuneable properties result from changes in the nanoparticle lattice

[1]State Key Lab of Metal Matrix Composites, School of Materials Science and Engineering, Shanghai Jiao Tong University, Shanghai 200240, China. [2]BASF SE, Dispersions & Resins, Carl-Bosch-Strasse 38, Ludwigshafen/Rhein 67056, Germany. [3]Department of Physics, Prifysgol Aberystwyth University, Wales SY23 3BZ, UK. [4]Department of Electrical Engineering, KU Leuven, Leuven B3001, Belgium. [5]Shanghai Synchrotron Radiation Facility, 201204 Shanghai, China. [6]School of Mechanical Engineering Sciences, University of Surrey, Guildford GU2 7XH, UK. [7]Institute of Forming Technology and Equipment, School of Materials Science and Engineering, Shanghai Jiao Tong University, 200240 Shanghai, China. [8]Department of Physics, University of Cambridge, JJ Thomson Ave, Cambridge CB3 0HE, UK. ✉e-mail: jjb12@cam.ac.uk; txfan@sjtu.edu.cn; zhaoqibin@sjtu.edu.cn

structures upon mechanical deformation. When the volume fraction of the particles is extremely low, the deformation behaviours of the material and the change in the spatial arrangement of the particles mainly depend on the polymer matrix[10]. As the particles become more densely packed, interactions between hard particles become non-negligible (Fig. 1a). This is especially true when the particles are packed in ordered 3D lattice structures[11]. Uniaxial stretching of the material can result in complicated deformation behaviours that involve inter-particle interactions influenced by the lattice structure, particle-matrix interactions, and the deformation of the matrix itself. Compared to densely packed particles in random structures, which may show iso-tropic deformation insensitive to the stretching direction, the sym-metry of the ordered 3D lattice may impart strong anisotropy that depends on the stretching direction. Currently, there is still a lack of theories and experimental progress to fully unveil the deformation mechanism.

Revealing these deformation mechanisms also has broader implications. Colloidal particle assemblies are considered close ana-logs of atomic lattices in metals and are easier to study due to their much larger structural scale[12]. There has been extensive research on crystallization[13], defect growth[14], and phase changes[15], known as solid-solid transitions, in colloidal particle lattices, comparing them with those in atomic systems[16]. However, pure colloidal particle lattices usually cannot withstand much strain, whereas many metals exhibit high ductility[17]. Therefore, for a long time, there has been a question of whether nanoparticle lattices embedded in elastic polymers can serve as analogs for studying the mechanical deformation of ductile metals[18]. This question remains open until the deformation mechanism of elastic nanoparticle lattices is clearly understood. Furthermore, in the

frontiers of photonic research, many materials use particles of various shapes to assemble into lattice structures for novel properties[1,3,19]. For instance, topological photonic crystals (PCs) rely heavily on structural symmetry to display various optical effects such as the bound states in the continuum[20,21]. These effects can be dynamically tuned by rever-sibly breaking the original symmetry in structure deformation, resulting in controlled switching between multiple states[22]. While tuneable topological phase transitions have been demonstrated in stretchable 2D photonic crystals[22], achieving this in 3D photonic crystals remains a challenge since the prerequisite is to understand how the 3D lattices change.

The optical properties and bandgap structures of fixed-structure 3D photonic crystals have been extensively studied[23–26], yet the evo-lution of these properties under strain have not been fully unveiled. A wide variety of stretchable 3D PCs have been developed over the past decades[27–29], but studies focusing on their 3D lattice deformations and the resultant optical properties are still lacking. To achieve reasonable elasticity, these materials typically require an approximately 50%:50% volume ratio between rigid particles and the elastic matrix[28,30]. Con-sequently, despite the particles often being arranged in close-packing lattices such as face-centred cubic (*fcc*), hexagonal close-packing (*hcp*), and random-stacked hexagonal planes (*r*-hcp), these materials are often referred to as non-close-packed responsive photonic crystals[30,31]. This designation is because the particles do not directly touch, which otherwise would result in a particle filling fraction of ~74%, as seen in brittle synthetic opals. To avoid ambiguity, 'close-packed' in the fol-lowing context specifically refers to the type of lattice structure, not the overall packing density. Due to limitations in assembly mechan-isms and fabrication techniques for stretchable 3D photonic crystals

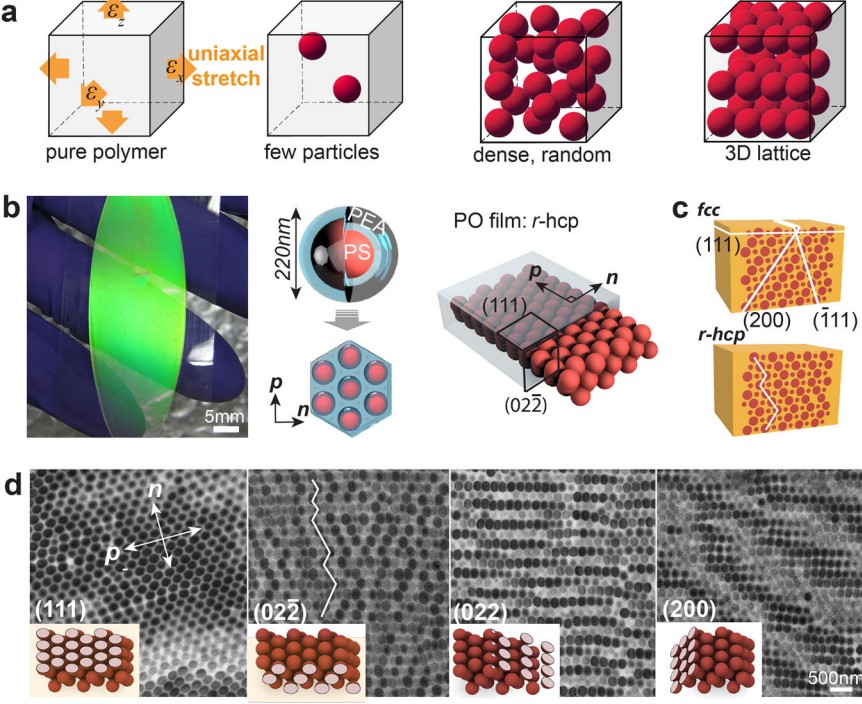

**Fig. 1 | Different structural arrangements of particles in polymer matrix and the 3D structure of polymer opal. a** Uniaxial stretching of pure polymers, parti-cles embedded in polymers at low volume fractions, particles in polymers at high volume fractions but packed in random structures, particles in polymers at high volume fractions and packed in 3D ordered lattices. **b** A green PO sample com-prised of 220 nm PS-PEA core-shell spheres. Illustrations on the right show the assembly of core-shell spheres into a film of a close-packing structure, where the (111) plane runs parallel to the film surface, and the (02$\bar{2}$) plane is the cross-section in the width direction. The *n*- and *p*-directions are indicated in the hexagonal

arrangement. **c** Illustration on the differences in the orientations of ($\bar{1}$11) and (200) planes between *fcc* and *r*-hcp lattices, the (111) plane is parallel to the surface, the revealed cross-section in this illustration is the (02$\bar{2}$) plane. **d** TEM images of in-plane packing structures of four different sets of lattice planes, the insets show the ideal orientations and structures of corresponding planes in a defect-free lattice. The vertical direction in the (02$\bar{2}$) image is the thickness direction, (111) planes are in the horizontal direction and stacked vertically. The stacking faults twist the orientations of tilted planes to opposite directions as shown with the white bro-ken line.

that require at least centimetre scale films[13,28,32], other lattice structures are possible but rare[33,34]. Particles in these materials are stacked in dozens to hundreds of layers parallel to the material surface, known as the (111) plane in *fcc* coordinates. Within each layer, they are arranged in a hexagonal pattern. These arrangements create various lattice planes tilted from the surface, such as (200) and ($\bar{1}$11) planes. Studies on non-stretchable synthetic opals have highlighted the main pseudogap or stopband from the (111) planes and bandgaps from tilted planes when incident light is angled at high values of ~50°–60° from the normal direction[24,25], leading to van Hove singularities[23]. The singularities are strongly related with symmetry of the 1st Brillouin zone, where it involves the complex interplay of multiple Bragg planes.

Most research on stretchable 3D-PCs has focused on the shift of the main pseudogap from (111) planes[32], with little reported on the evolution of photonic bandgaps of other lattice planes[18]. However, stretching the material uniaxially will undoubtedly alter the 3D lattice and reshape the symmetry of its entire photonic bandgap structure, necessitating further investigation. Moreover, despite efforts to explore the 3D lattice deformation of responsive photonic crystals upon stretching, convincing theoretical frameworks to elucidate experimental results are notably absent[35]. The complex 3D lattice deformation is often simplified to changes in the spacing between layers in a multilayer structure composed of (111) planes[13,36]. However, it is obvious that the hexagonal arrangement of particles within the (111) layers varies when the material is stretched in different directions. This leads to a contradiction: the configuration within each layer depends on the direction of stretching, yet the overall deformation across multiple layers fails to account for the direction of planar stretching. While treating the stretching of 3D-PCs as a thinning process of a 1D multilayer structure provides a straightforward explanation for the blueshift of the main (111) pseudogap, this approach does not offer insights into the behaviour of various other planes. A comprehensive 3D model is crucial to reveal the deformation of various lattice structures and the resultant optical effects.

In this study, we present our findings on archetypical stretchable 3D photonic crystals composed of polymer beads embedded by grafting into a viscoelastic medium. We demonstrate that the rearrangement of spheres within these elastic composites is governed by the lattice structure and driven by the surrounding matrix, a physical process that can be accurately simulated with analytical models. Our models identify two distinct deformation pathways when the material is stretched in orthogonal directions: one leads to an immediate transformation from *fcc* to triclinic lattice, continuing to deform with further stretching, while the other involves more complex intermediates between two *fcc* structures. The model predictions are supported by extensive mechanical, optical and structure characterizations from 0-200% strain. The 3D mechanical models allow us to explore the strain-tuned interplay and evolutions of multiple photonic bandgaps from different lattice planes. We clarify distinct shifts of the main pseudogap of (111) planes when stretching the material in different directions. Notably, as the photonic bandgaps from tilted (200) and ($\bar{1}$11) planes shift to lower energies and the (111) pseudogap shifts to higher energies, the van Hove singularity occurs at incident angles much closer to normal incidence. Stretching induces significant deformation of the Brillouin zone. Above a threshold of 40% strain, the 1st Brillouin zone excludes the (111) Bragg plane and mainly contains (200) and ($\bar{1}$11) Bragg planes. We report the merging of different symmetry points and the formation of new symmetries. A possible optical singularity providing different sets of tilted planes is revealed, which always happens at normal direction but shifts in wavelength. The stretched materials display retro-reflecting colours from the direct escape of light from stopbands of the tilted planes. Furthermore, we explicitly illustrate how light escapes the material in different ways, as tuned by strain, displaying varied colour effects. This work lays the ground for investigating strain-induced structure changes of nanoparticle lattices in elastic materials. It bridges existing gaps in material science and photonic research and can provide a pivotal physical exemplar system to guide cutting-edge research in topological photonics. Moreover, the demonstrated tuneable optical properties may pave new avenues for 3D photonic crystals in emission control[9], optical camouflage[37], and augmented reality devices[38].

## Results

### Original 3D packing structure

The materials we use are termed polymer opals (POs), and exhibit excellent stretchability and reversible colour changes upon strain. The POs are fabricated using polymeric core-shell spheres comprising a hard polystyrene (PS) core and a soft poly (ethyl acrylate) (PEA) shell (Fig. 1b, Fig. S1). These spheres are shear-assembled into a classic close-packing structure[7], wherein the PEA shell melts and fills all the interstices. The overall size of the core-shell sphere used in our experiments is ~220 nm, unless otherwise stated. The PEA matrix constitutes about 55% of the total volume, ensuring that the PS spheres do not directly touch each other in the resultant photonic crystal film after assembly, thus achieving a critical balance between optical performance and elasticity. Like most stretchable 3D photonic crystals, POs have layers of hexagonally arranged PS spheres aligned parallel to the surface (Fig. 1b). The typical thickness of PO films in this study is ~80μm, comprising 300-500 such layers. Due to the structural similarities between *fcc*, *hcp* and *r*-hcp close-packing lattices, for the convenience of discussion, we consistently use the Miller indices of the *fcc* lattice to refer to the same structures in close-packing lattices (Fig. S2). Therefore, the layers parallel to the surface are the (111) planes. Owing to their hexagonal arrangement, there are two primary directions within the (111) planes: the *p*-direction along the rows of spheres, and the *n*-direction perpendicular to it (Fig. 1b).

By slicing a cross-section in the *n*-direction (Fig. 1b), transmission electron microscope (TEM) images clearly reveal how the (111) layers are stacked in the thickness direction (Fig. 1d). The exposed cross-section is the (02$\bar{2}$) plane in the 3D lattice. In an ideal *fcc* lattice without stacking faults between the (111) planes, spheres will align in straight lines that are tilted from the surface (Fig. 1c), each representing a specific lattice plane such as (200) and ($\bar{1}$11). In POs, the straight lines are often broken and twisted to opposite directions, manifesting large quantities of stacking faults between the (111) layers[39], which result in the frequent switching of the orientations of the (200) and ($\bar{1}$11) with each stacking fault, typical of a *r*-hcp lattice (Fig. S2). Since *fcc*, *hcp* and *r*-hcp lattice are convertible to each other by varying the stacking faults, *r*-hcp can be considered a highly intercalated mixture of *fcc* and *hcp*.

### Anisotropic deformation

We first investigate the strain-induced colour changes at normal incidence (Fig. 2a). Before stretching, POs show a green colour at ~560 nm at normal incidence (Fig. 2b, c, Fig. S3), which results from the Bragg reflection of (111) planes, also known as the main pseudogap, determined by $2nd_{111} = \lambda$. Here, *n* is ~1.54 from the mean refractive index of PO, $d_{111}$ is the spacing between the (111) layers and $\lambda$ is the reflection wavelength at normal incidence. Not surprisingly, when stretched, the Bragg reflection shifts to shorter wavelengths as the inter-layer distance decreases. However, we find colour shifting in *n*-stretching is almost twice as fast as that in *p*-stretching (Fig. 2c, d). Spectral characterizations reveal that 40% strain ($\varepsilon_x$) in *n*-stretching induces a 19% ($\varepsilon_{refl}$) decrease of the reflection peak wavelength, compared to 11% in *p*-stretching, indicating distinct colour-strain sensitivities indicated by $\varepsilon_{refl}/\varepsilon_x$. This agrees with previous work on these POs which showed such anisotropies[18]. This finding is critically important for responsive 3D photonic crystals, especially those intended for use as strain sensors. Previously, responsive 3D photonic crystals were often treated as 1D multilayer structures, in which the optical shift of the (111)

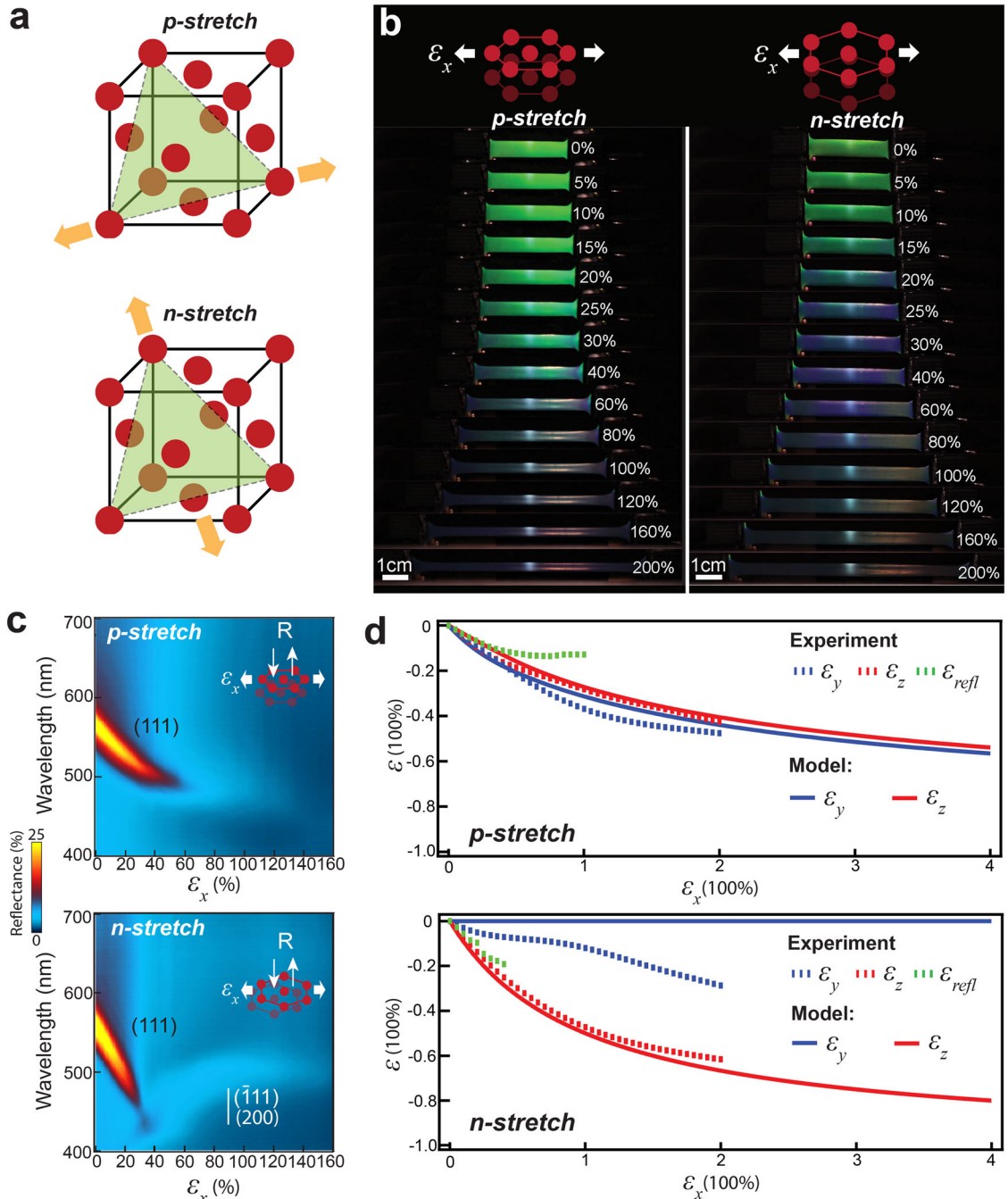

**Fig. 2 | Anisotropies in the deformation and optical shifts of POs in uniaxial stretching. a** Illustration of the *p*- and *n*-stretching of the 3D lattice of PO, the spheres in the illustration do not represent the actual size. **b** Stretching dynamics of POs in *n* and *p*-directions, with the central bright spot arising from microscope objective illumination used for in-situ thickness monitoring, stretching strain is indicates as $\varepsilon_x$. Colour shifts and changes in the specimen dimensions are revealed in the images taken normal to the specimen surface during in-situ stretching. **c** Bright-field reflectance spectra of *p*- and *n*-stretched POs at normal incidence during stretching. **d** Measured and model predicted deformation strains in *n*- and *p*-stretching, $\varepsilon_y$ is width strain, $\varepsilon_z$ is thickness strain, $\varepsilon_{refl}$ is the normal incidence (111) plane Bragg reflection peak wavelength shift as shown in the spectra in (**c**).

pseudogap was considered irrelevant to the direction of stretching. Therefore, the stretching direction is often not specified in studies, leading to confusion about why materials with extremely similar structures display different color-strain sensitivities. Our finding provides an explanation for this phenomenon.

The colour shifts originate from distinct changes in the PO's dimensions when stretched in different directions. While the pure polymer material and POs with highly disordered structures exhibit almost equal changes in thickness and width upon stretching (Fig. S4),

POs with ordered lattices display strong anisotropy in deformation (Fig. 2d, Fig. S5). Our results indicate that, compared to *p*-stretching, which results in similar width ($\varepsilon_y$) and thickness ($\varepsilon_z$) deformation, *n*-stretching leads to a more pronounced decrease in thickness relative to width. This observation suggests a strong relationship between the stretching direction and the symmetry of the 3D lattice. Additionally, we find clear deviations between the optical shift $\varepsilon_{refl}$ of the (111) Bragg reflection and the measured actual thickness change $\varepsilon_z$. We attribute these deviations to structural disorders, noting that they can be

minimized with significantly improved structural order (Fig. S6). Conventionally, researchers have often used the optical shift $\varepsilon_{refl}$ to calculate the thickness change of the material, under the assumption that $\varepsilon_{refl}$ is determined solely by the distance between the (111) layers[13,38]. However, the pronounced presence of disorders in most responsive 3D photonic crystals, coupled with the observed deviations between $\varepsilon_{refl}$ and $\varepsilon_z$, which become more pronounced at larger strains, suggests that the optical shift should be considered a qualitative rather than a quantitative indicator of actual thickness changes.

## 3D Lattice transformation in *n*-stretching

The pronounced differences in optical shifts and macroscale deformations strongly indicate distinct lattice transformation processes during *n*- and *p*-stretching, as discussed in[18]. However, a more crucial task is to reveal how exactly the lattice changes. Due to the limitations of available structural characterization techniques for these materials, and the disturbances caused by structural disorders in interpreting experimental results, it is almost impossible to clearly unveil the complete deformation process without guidance from solid quantitative theoretical models. The main challenge lies in identifying what drives this deformation process[11,18]. Once the fundamental principles are established, solutions can become quite straightforward. Here, we present our work, considering the deformation process as a joint result of interactions between spheres in the lattice structure and the strong influence of the matrix material.

We use a general tetrahedron unit cell to model the deformation process (Fig. 3a). This unit cell contains rigid PS spheres at the vertices and the elastic PEA matrix in the interstices, applicable to all types of close-packing lattices. We first establish the relations of width and thickness strains to the stretching strain ($\varepsilon_y - \varepsilon_x$, $\varepsilon_z - \varepsilon_x$), which require at least two boundary conditions (Supplementary Discussion 1). The constant volume of the unit cell due to incompressibility of the provides the first boundary condition. The second condition lies in the tetrahedron geometry related to the distance between the apex sphere and the three base spheres in stretching. In plasticity theories of metals or solid-solid transitions of colloidal crystals[15,40,41], this distance keeps almost constant due to interparticle bonding. However, maintaining a constant interparticle distance will lead to an accelerated decrease in thickness (Fig. S7), as shown in previous studies[11], contradicting our experimental observations. Our approach is to first leave the question open and then calculate whether it is a constant or variable distance between the particles with our model. The calculations reveal a dynamical interparticle distance rather than a fixed value (Fig. S8), highlighting the elastic nature of the material. The model-calculated stretching-induced strains, in both thickness and width, as well as the dynamic interparticle distance, align well with experimental results (Fig. 2d). It's evident that in *n*-stretching, the rows of spheres aligned in the width direction strongly resist decreases in width. Thus, the slight decrease in width observed in the experiment is likely due to the compression of the matrix material and disruptions to the rows of spheres resulting from increasing stress and structural disorder (Fig. S9)[42].

The model accurately predicts the exact positions of each sphere and the resultant lattice structures at different strains. The predicted structural evolutions are illustrated using an *fcc* with a rhombohedral primitive cell as the initial lattice for convenience (Fig. 3b–f). In *n*-

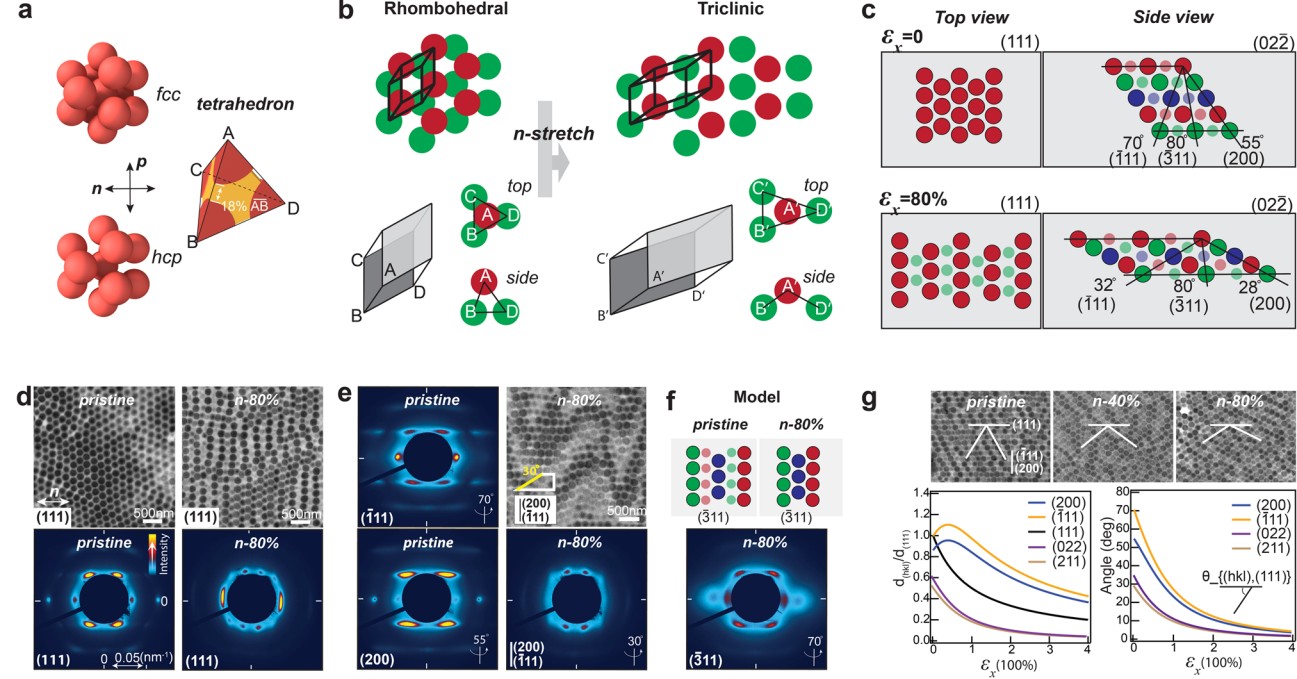

**Fig. 3 | 3D lattice transformations in *n*-stretching. a** Structure of the tetrahedron unit PS spheres are shown in red, yellow shows the PEA matrix. **b** Transformation of the original close-packing lattice to triclinic lattices in *n*-stretching. Spheres in two neighbouring (111) layers are shown in red and green, respectively. The original close-packing lattice can be represented using a rhombohedral primitive cell (thick black). **c** Predicted lattice structures pre and post 80% *n*-stretching using an initial *fcc* lattice. Three repeating (111) layers are shown in red, green, and blue colours, respectively. Top view shows the sphere arrangement within the (111) plane, side view is the cross-section in width direction, showing the sphere arrangements within (02$\bar{2}$), orientations of different tilted planes are shown in the side view. **d** Top, TEM images of (111) in-plane structures pre and post-*n*-stretching, bottom,

corresponding SAXS patterns normal to the (111) plane. **e** Left, SAXS patterns normal to ($\bar{1}$11) and (200) planes pre stretching, right top, TEM image of the structure revealed from cutting an 80% *n*-stretched sample at 30° from the surface, showing mixed ($\bar{1}$11) and (200) in-plane structures, right bottom, the corresponding SAXS pattern normal to the revealed structure. **f** Top, model predicted spheres arrangements within the ($\bar{3}$11) plane pre and post 80% *n*-stretching, bottom, SAXS pattern obtained at 70°, near the predicted ($\bar{3}$11) plane orientation of 80°. **g** Top, TEM images of (02$\bar{2}$) in-plane structure in *n*-stretching, relative orientations of tilted planes toward (111) are shown in while lines, bottom, model predicted evolutions of the lattice distance and orientations of various planes.

stretching, the rows of spheres within the (111) layers are stretched further apart (Fig. 3c), accompanied by a decrease in the (111) interlayer spacing, resulting in a transformation from *fcc* to triclinic lattices. Planes such as ($\bar{1}$11), (200), (022) and (211), originally titled from the surface, continuously rotate towards the (111) planes, as shown by the changes in their relative angles (Fig. 3c, g). The plane distances of ($\bar{1}$11) and (200) initially increase by a maximum of ~10% till ~40% strain then gradually decrease. These predictions match well with TEM and small-angle X-ray scattering (SAXS) characterization results (Fig. 3d–f). The model's accuracy is further substantiated by examining the ($\bar{3}$11) plane, which requires precise structure prediction across multiple layers. Notably, this plane maintains the same orientation during *n*-stretching.

As previously discussed, stacking faults between the (111) planes cause slanted planes such as ($\bar{1}$11) and (200) to twist into mirror orientations. We denote these mirror orientations as left- and right-tilting, with both orientations present in these planes. Before stretching, the orientation difference between ($\bar{1}$11) and (200) is only ~15°, which diminishes in *n*-stretching to approximately 4° at 80% strain. As a result, structure characterizations of any of these planes often display mixed sphere arrangements of both (Fig. 3e).

## Optical transitions in *n*-stretching

We observe striking retro-reflective colours in *n*-stretched POs (Fig. 4a, Movie S1), evident in samples made from different spheres (Fig. S10). Previous research has shown retro-reflection colours in stretched opals using the diffraction of surface spheres[43], similar to the rainbow diffraction colours of colloidal monolayers[44]. Surface diffraction can be eliminated by filling the gaps between the protruding spheres with a refractive index matching oil, yet the colours in our *n*-stretched samples remain (Fig. S9, Movie S3). Here, we offer a new mechanism and unveil the rich mechano-chromic properties resulting from the lattice deformations induced by *n*-stretching.

We primarily focus on the optical performance from the main pseudogap of the (111) planes and stopbands of the (200) and ($\bar{1}$11)

planes. Considering that the bandgap diagrams predict light that is allowed or forbidden to propagate within the material, it is equally important to reveal how light escapes and reaches the detector. To this end, we use a Bragg reflection coupled diffraction (BCD) model based on multiple diffraction theories (Fig. 4c, Fig. S11)[45,46]. Details of the calculations are shown in Supplementary Discussion 2. In this model, both the incident light and the Bragg reflections from the tilting planes inside the material are diffracted by the (111) planes, which agrees with the central ideas in previous studies with synthetic opals[23,25]. The spectra are calculated using the dispersion of their Bragg reflections as revealed with the Brillouin zone (Fig. 5a). There are more rigorous theoretical approaches that describe the optical properties using density of states and photonic bandgap calculations, however, since we primarily focus on the wavelength and angle shifts, the simple model provides satisfying results. There is only a notable deviation between our model and the more rigorous methods at positions where van Hove singularities happen, producing an anti-crossing effect between spectra of Bragg planes. Here the positive side to allow the spectra to cross in our model is that the positions of the crossing can clearly indicate strain-induced evolution of the symmetry points in the Brillouin zone, such as the U and K points. Considering the special structure symmetry of the *r*-hcp lattice, we construct a reduced 1st Brillouin zone using the (111), (200), and ($\bar{1}$11) Bragg planes (Fig. 5a). Notably, the ($\bar{1}$11) and (200) Bragg planes present in both sides. Therefore, the symmetry points U at the crossing edge between (111) and (200) planes, and K between (111) and ($\bar{1}$11) planes are equivalent. The Γ-L direction is normal to the (111) planes.

According to the model, when incident light is directed onto the material (Fig. 4c), it enters the interior through refraction at an angle determined by Snell's law. As it propagates through the hundreds of layers of spheres, it induces diffraction by the (111) planes. The resultant zero-order diffraction corresponds to the Bragg reflection of the (111) planes (PR in Fig. 4c), exiting the material in the specular reflection direction. Its first-order diffraction (PD in Fig. 4c) occurs in different directions, depending on the wavelength. The incident light also

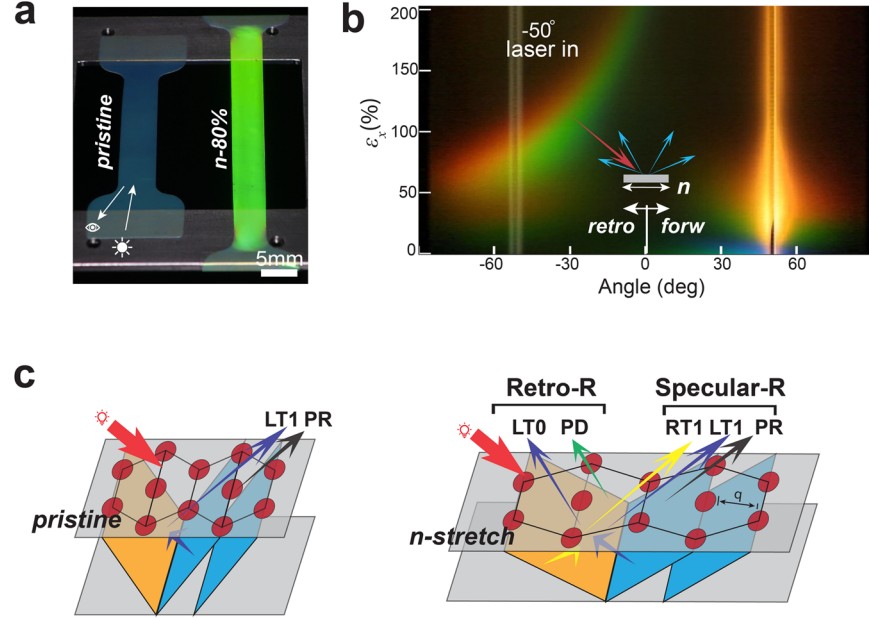

**Fig. 4 | Optical transitions in *n*-stretching. a** Retro-reflection colours of POs pre and post *n*-stretching. **b** Angular scattering colour of a continuously *n*-stretched sample captured at $\varphi_1 = -50°$ incident angle, incident light is a supercontinuum white light laser, colours are true colours by imaging the screen. **c**, The reflections in the BCD model pre and post *n*-stretching, the incident light coming from the left along *n*-direction is shown in a thick red arrow, (111) planes are indicated in grey, the

tilted planes such as ($\bar{1}$11) and (200) are shown in blue as left-titling and orange as right-tilting, q is the distance between two rows of spheres, PR and PD are the zero- and first- order diffraction of incident light, respectively, LT0 and LT1 are the zero- and first-order diffractions of the Bragg reflection from left- tilting planes, respectively, RT0 and RT1 are those of the right-tilting planes.

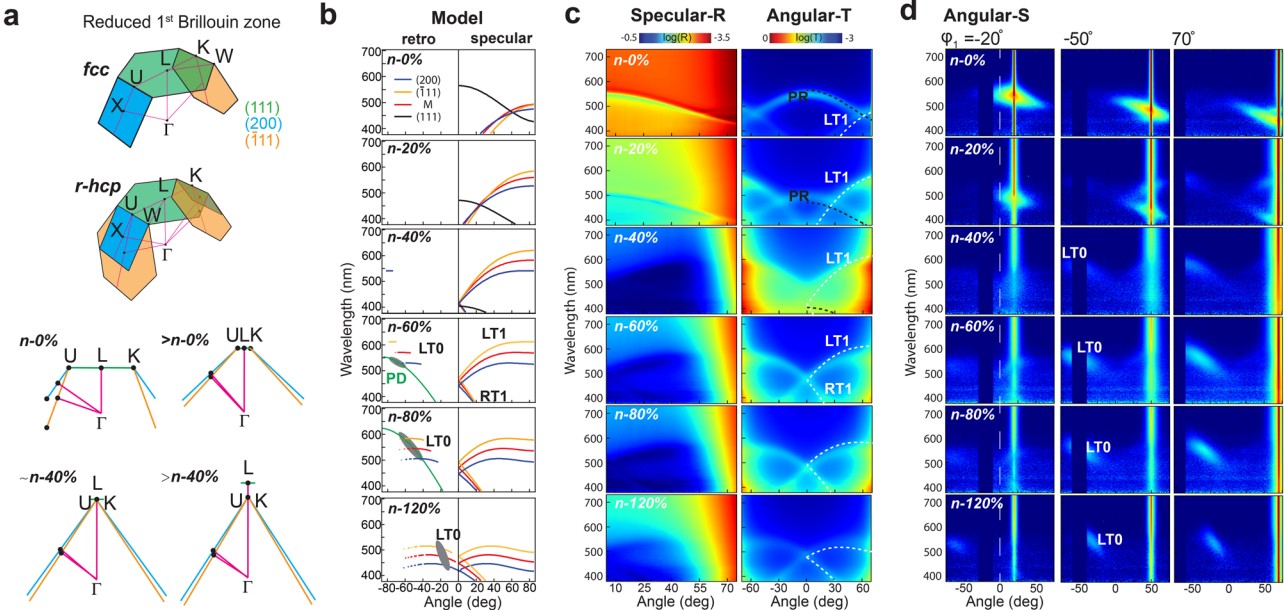

**Fig. 5 | Model predictions and experimental spectra of the optical transitions in *n*-stretching. a** Deformation of the reduced 1st Brillouin zone in *n*-stretching, Brillouin zone planes of (111), ($\bar{1}$11) and (200) planes are shown in different colours, U is the symmetry point at the edge between (111) and (200), K is the symmetry point at the edge of (111) and ($\bar{1}$11) planes. In the Brillouin zone of the *r*-hcp lattice, U and K are equivalent. **b** Model calculated reflection spectra of selected planes in specular and retro-reflection directions at varying incident angle. The first-order diffraction spectra of incident light (PD) at $\varphi_1 = -50°$ are highlighted in green for 60% and 80% *n*-stretching. Wavelengths of retro-Bragg reflection LT0 at $\varphi_1 = -50°$ are determined by the crossing points between PD and LT0, all possible retro-reflections at −50° incidence are indicated in dark grey ellipses. LT1 is the reflection spectra of the left-tilting planes, RT1 is that of the right-tilting planes, they meet and the turning point of 0°. A set of 'M' planes is constructed using the average tilting angle and plane distance of ($\bar{1}$11) and (200), to provide a reference on the effects of their orientation and plane distance differences. **c** Measured gonio-specular reflection and transmission spectra of *n*-stretched samples. Model prediction of (111) reflection PR is shown in black dashed line, while dashed lines use ($\bar{1}$11) plane's predicted LT1 and RT1 reflection. **d** Angular distribution of scattering spectra for *n*-stretched samples varying in incidence angle and strain, LT0, the Bragg reflected retro-reflection from left-tilting planes, appears in ellipses as predicted.

triggers Bragg reflections of the ($\bar{1}$11) and (200) planes. Since these planes are present in both left- and right-tilting orientations, we refer to them as 'LT' and 'RT' planes, respectively. When their Bragg reflections reach the air-PO interface, they either directly escape into the air through refraction or return to the interior due to total internal reflection above a critical angle of approximately 40°. The constrained light propagates within the material, is known as 'guided resonances.' During this, a portion of the Bragg reflections can always exit the surface due to first-order diffraction by the (111) grating structure, known as 'leaked' or 'extracted' light (LT1 and RT1, Fig. 4c). In the current regime, when light is incident from the left, the Bragg reflection of RT planes can only escape via diffraction, with the direction coinciding with the Bragg reflection direction of the (111) planes. Due to the small refractive index contrast ($\Delta n = 0.1$) between the sphere and the matrix, the measured optical signals are results from hundreds of layers. The influences of stacking faults and stretching-induced deformations often drive the (200) and ($\bar{1}$11) planes to become close in their lattice distances and tilting angles (Fig. S12), which may result in merged spectra from their collective reflections.

The strain-induced optical properties are revealed with angle-resolved spectroscopy. Before *n*-stretching, PO exhibits the typical angle-resolved spectroscopy characteristic of an opal photonic crystal (Fig. 5b, c). The initial coloration of the material primarily depends on the main pseudogap of the (111) planes. The Bragg reflection from the (111) planes occurs in the forward specular reflection direction and shifts to shorter wavelengths as the incident angle increases (L-U or L-K). The van Hove singularity effect is observed at approximately 55°, where it fulfils Bragg conditions of left-tilting ($\bar{1}$11), (200) and (111) planes at U or K point (Fig. 5a). At this stage, Bragg reflections from the ($\bar{1}$11) and (200) planes cannot directly escape due to total internal reflection; otherwise, they would appear in the retro-reflection direction of the incident light.

The observed signals are their first-order diffraction and always aligns with the (111) Bragg reflection (Supplementary Discussions 2). As the 3D lattice deforms from *n*-stretching, the (111) pseudogap quickly shifts to shorter wavelengths, and the Bragg reflections from the ($\bar{1}$11) and (200) planes shift to longer wavelengths (Fig. 5b, c, Fig. S12, S13). In the reduced 1st Brillouin zone, these are indicated in the elongation of the Γ-L dimension and the shifting of U or K point toward L point. The wavelength at which the van Hove singularity occurs remains relatively unchanged until 20% strain, after which there are rapid decreases with further stretching. The angle at which it can be observed reduces to approximately 0°, normal to the surface, near 40% strain, where U (left) and K (right) points coincide with L point, merging as a new single symmetry point (Fig. 5a).

From near 40% strain, new optical effects arise from the evolving structural symmetries. Due to the continuous tilting of the ($\bar{1}$11) and (200) planes, their Bragg reflections finally overcome total internal reflection and directly escape the surface through refraction. Initially, they appear as retro-reflection colours visible only from directions parallel to the surface (Fig. 5b, d). As the strain increases, their visible directions tune toward the normal direction (Fig. 5d, Fig. S14). According to our model calculations, the direction of the retro-reflection colours coincides with the first-order diffraction of incident light by the (111) planes, which visually appears as rainbow colours (Fig. 4b, Fig. S15, Movie S2). However, they are distinctly identified through spectroscopy. The retro-Bragg reflections from hundreds of layers of ($\bar{1}$11) and (200) planes merge into a strong ellipse-shaped spot in the scattering spectra (Fig. 5d). This effect is more clearly revealed by using samples with improved structural order (Fig. S16). Meanwhile, both the first-order diffraction of the Bragg reflection from the LT planes and that from the RT planes present in reflection and align in the forward specular reflection direction. While they display distinct

angle-dependent wavelength shifts, their spectra always meet at 0° due to the stretching-induced new structure symmetry. As shown in the reduced 1st Brillouin zone (Fig. 5a), above 40% strain, the L point of the (111) Bragg plane becomes further distanced from the Γ point, and the (111) Bragg plane is no longer in the 1st Brillouin zone, leaving only the RT and LT ($\bar{1}11$) and (200) Bragg planes that always intersect at the merged symmetry point of U and K, normal to the (111) planes. Thus, *n*-stretching may cause the optical effects related with singularities to transition through three distinct states: at the symmetry point between (111) and either RT or LT planes, at the merged point of (111), RT, and LT planes, and at the symmetry point between RT and LT planes, respectively.

### 3D Lattice transformation and optical transitions in *p*-stretching

We further reveal the lattice deformation from *p*-stretching. The process is modelled using three boundary conditions (Supplementary Discussion 1): (i) volume preservation, (ii) equivalency of (111) and ($\bar{1}11$) planes in *p*-stretching, and (iii) geometric relations within the unit cell. We demonstrate the lattice transformations again using the tetrahedron unit cell and *fcc* lattice (Fig. 6a). Here, the unit cell's base represents the (111) plane, while the facet 'BBC' corresponds to the ($\bar{1}11$) plane.

*P*-stretching induces a dual-phase lattice transformation marked by 42% strain. Initially, the increasing separation of spheres 'B-C' along the *p*-direction drives sphere 'D' towards 'B-C' within the (111) plane and sphere 'A' towards 'B-C' within the ($\bar{1}11$) plane, keeping a constant distance between 'A' and 'D'. This results in a marginally faster

reduction in width than thickness, corroborated by both model and experimental data (Fig. 2d). At 42% strain, the spheres arrange into same square patterns in both the cross-sections and surface, each sphere maintaining an equal interparticle distance with its twelve neighbours (Fig. 6a, b). The resultant lattice is effectively a rotational translation of the original *fcc* lattice, exposing its (200) plane to the surface. Beyond 42% strain, this configuration, resistant to further *p*-stretching, is disrupted by misaligned spheres, leading to defect-induced equal strain changes in width and thickness. This may explain the more pronounced structure disorder in *p*-stretching than *n*-stretching. In the second stage, it is difficult to classify the resultant structure as a monolithic lattice, but the lattice's indicative configuration can still be estimated using model calculations. By 80% strain, it morphs into a new structure mimicking its pre-stretching lattice but with switched *p*- and *n*-directions and altered lattice scales (Fig. 6c). This transformation, despite significant structural disorder, is supported by TEM and SAXS characterization results (Fig. 6b, c). In the new lattice, the ($02\bar{2}$) planes are reoriented to act as equivalents to the original ($\bar{1}11$) and (200) planes, suggesting a repetition of the *n*-stretching process in this further *p*-stretching extension beyond 80%.

The deformed lattices in *p*-stretching produce various new lattice symmetries that may result in rich interesting optical phenomena. However, due to constraints in orientation angle and plane distance, these signals are often in short wavelengths that are beyond our characterization methods that focusing on the visible range. Therefore, the PO sample under *p*-stretching primarily show

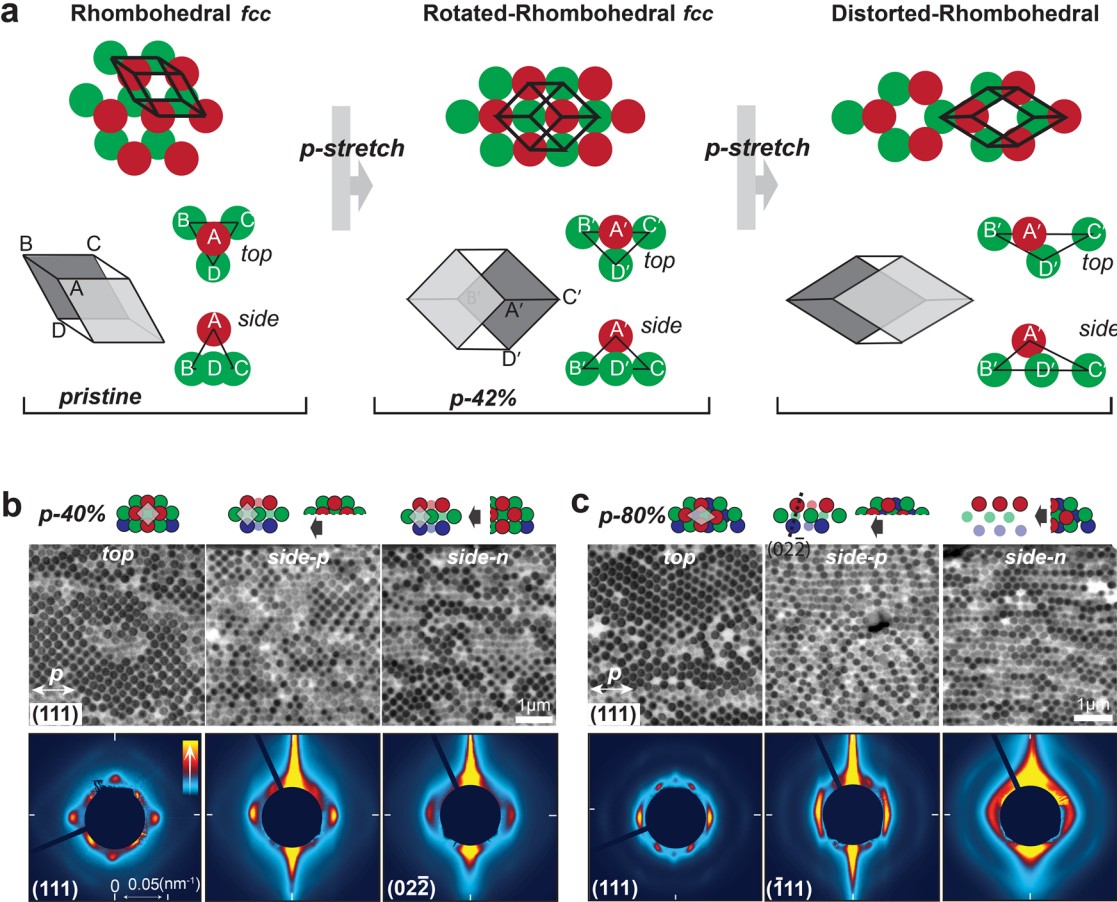

**Fig. 6 | 3D lattice transformations in *p*-stretching. a** Dual-stage 3D lattice transformation process in *p*-stretching, this process is shown with the rhombohedral and tetrahedron unit cells, *fcc* is adopted as the initial lattice structure for convenience. **b** Top, model predicted surface and cross-section structures at 40% *p*-stretching, the TEM images and SAXS patterns below show corresponding structures obtained in experiments. **c** Model predicted structures and experimental results at 80% *p*-stretching, the black dashed line in the illustration indicates the orientation of ($02\bar{2}$) plane.

structural colours from Bragg reflection of the (111) planes, but beyond 160% strain it may exhibit retro-reflection colours similar to $n$-stretched POs above 40% strain. This is confirmed by experimental results (Fig. S17), but these colours are rather faint due to significant structural disorder.

## Light tuning with strain

The flexible manipulation of light with strain is more effectively demonstrated using samples with larger spheres of ~250 nm When the sample surface is illuminated at normal incidence with a 532 nm monochromatic laser, weak diffuse light is induced within the material due to scattering. This diffuse light is enhanced in directions that meet the Bragg conditions (Fig. 7a), resulting in lines known as the Kossel pattern[47]. We demonstrate that both $p$- and $n$-stretching can tune the beams to rapidly scan over wide angles. Additionally, we showcase the tuneable spreading of structural colours over large areas during $n$-stretching (Fig. 7b). The sample is attached to an elastic transparent tape and illuminated with a supercontinuum white light laser. At normal incidence, the Bragg reflection from tilting planes is subject to total internal reflection until approximately 130% strain. Below this threshold, it propagates within the material as the 'guided resonance'[48]. Its energy is gradually extracted by diffraction of the grating structure, escaping the material surface as the 'leaked' light opposing to incident light. These strain-tuned optical effects from the tilted planes rather than the (111) planes can be further combined with diverse optical mechanisms such as photonic band edge effects for diverse applications such as photoluminescence control[49].

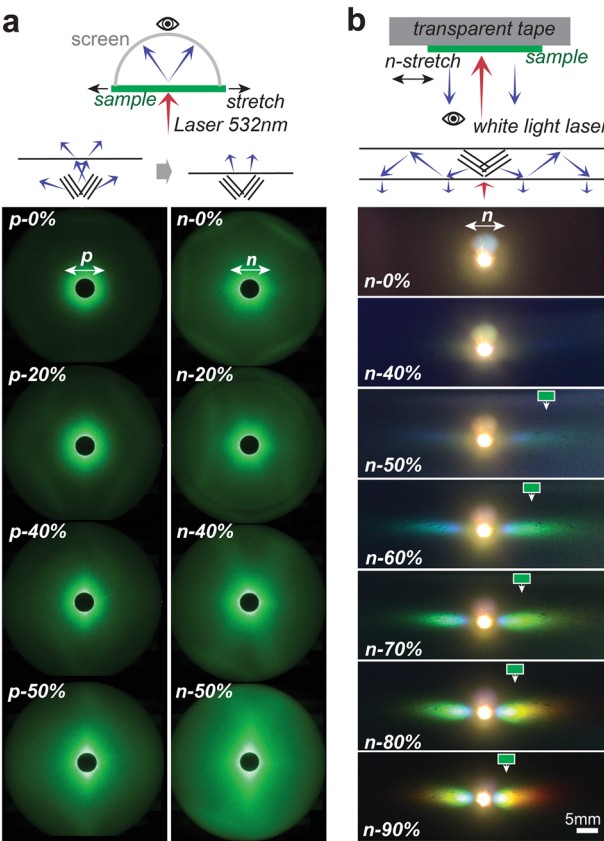

**Fig. 7 | Dynamic tuning of the Bragg reflections from the tilting planes.**
a Dynamic evolution of the Kossel patterns with strain. **b** Tuning of the reflection colours using the guided mode and the multiple diffraction mechanism gives colours which are brighter over larger areas to the eye, since the overall brightness of the image is reduced to avoid the laser spot being too strong, the green label indicate the moving of the reflection light in lateral position.

## Discussion

This work provides general principles and mechanisms for understanding and predicting the deformation behaviours of 3D nanoparticle lattices in elastic materials. Similar models can be constructed for materials with various lattice structures by identifying the appropriate boundary conditions using the same approach. The stretching-induced deformation is neither facilitated by defects nor by slipping, as observed in ductile metals, nor does it result from the breaking of short-range interparticle interactions, as seen in colloidal lattices in liquids or air. Instead, it combines the elasticity of polymers with the control exerted by the regular spatial arrangement of rigid nanoparticles. These particles act as small domains within an elastic polymer matrix and are significantly stiffer than the surrounding material. Our results are applicable to materials with high concentrations of stiff domains packed in periodic configurations, which are not currently addressed by conventional theories of polymer or polymer composite deformation. Moreover, it overcomes the limitations of the commonly adopted 1D multilayer model for the structural and optical changes of stretchable photonic crystals, providing new insights by unveiling the complete 3D landscape. This establishes a strong foundation for their future development. The stretching-induced lattice transformations of photonic structures here shares many elements with the conserved properties of topological photonics under continuous deformation[21], potentially opening the way to robust new optical functionalities.

## Methods
### PO fabrication

Comprehensive details regarding the BIOS method and the synthesis of core-shell spheres can be found in our earlier publications on polymer opals. Following the BIOS process, PO films are typically encapsulated between two PET foils. These PET layers were subsequently removed, yielding freestanding PO films for subsequent experimentation. These films exhibited an average thickness of approximately 80μm. The $p$-direction and $n$-direction orientations of the PO film are determined during the BIOS process, with the $p$-direction aligned with the shearing direction. Dumbbell-shaped PO specimens, maintaining consistent dimensions, were prepared for tensile testing after UV cross-linking.

### Characterizations of mechanical and optical dynamics in continuous $p$ and $n$-direction stretching

A TCTS350-H tensile test stage was integrated onto a bespoke RM50M optical microscope for combined mechanical and reflectance assessments. The tensile stage was slightly modified so to eliminate back-reflection of transmitted light. This microscope was further coupled with a fibre-coupled Ocean Optics QE Pro spectrometer. In-situ characterizations consistently utilized a stretching speed of 0.1 mm/s to accommodate spectra and image acquisition. Optical microscopic images of the specimen's thickness cross-section during stretching were acquired using a 10X objective in darkfield mode, capturing the side view. Variations in surface colours throughout the procedure were documented with a Canon digital camera. When examining the surface during stretching, microscopic images were taken with a 5X objective (NA = 0.15) in brightfield mode integrated into an Infinity camera. Concurrently, reflectance spectra, normal to the surface, were recorded, spot size was ~300μm. Reference was taken with a silver mirror. Microscopic local strains were deduced by evaluating the relative displacements of defect markers within both cross-sectional and surface imagery, avoiding influence from interruptions such as slight surface bending at large strains. The cross-sectional images provided data on thickness strains and stretching strains, whereas surface images gave insights into width strains and stretching strains. Both thickness and width strains were subsequently reconciled using the stretching strain. Derived reflectance spectra were correlated with the microscopic strains, rather than the nominal strain as indicated by the tensile stage.

## Structure characterizations

PO specimens for SEM, TEM and SAXS characterizations were stretched to different strains and fixed using 3 M VHB tapes. Fixed specimens for TEM characterization were embedded using Struers EpoFix Kit, and microtomed into 80-100 nm thick slices at different specific tilt angles using Leica EM UC7FC7 at −80 to 120 °C. The slices were subsequently stained using $RuO_4$ vapour for 40 min, examined using a Thermalfisher Talos L120C G2 TEM under 120 kV. Montage TEM images of different areas of the sample were collimated and stitched automatically. Samples for SEM characterization were sputtered with gold for 45 s and examined using an FEI Sirion 200 SEM at 5 kV. A Varian Multimode and Dimension 3100 AFM were used in AFM characterization.

SAXS measurements were performed on the BL10U1 USAXS beamline at SSRF in Shanghai using a photon energy of 10 KeV. The diameter of the beam was ~300μm. An EIGER 4M 2D CCD detector was used to collect diffraction data. Background scans were taken by measuring scattering in air without samples. The $q$-range is calibrated using fibre diffraction of dry beef tendon.

## Scattering colour imaging

Specimens were subjected to stretching using the TCTS350-H tensile test stage for scattering imaging. Illumination was provided by an NKT Photonics SuperK Compact supercontinuum white light laser directed onto the sample. A partial cylindrical surface, possessing a 120° curvature, was positioned above the sample to serve as a screen. Continuous stretching at a rate of 0.1 mm/s was recorded using a Canon digital camera. This curved screen was preferred over a hemisphere, as it yielded richer data in the high-angle range, especially when the camera was stationed directly above. Angles within each video frame were calibrated to ensure a linear axis. By utilizing pixels from the central line (light in-out plane) of the scattering images at various strains, stacked scattering images were generated. Stretching strains indicated in scattering colour imaging uses the nominal strain from the stage.

## Angle-dependent spectral characterizations

PO specimens were stretched to different strains and attached with one side onto a VHB tape to prohibit relaxation. Angle-dependent specular reflection, transmission, and scattering properties were measured using a self-made goniometer. Two focal distance adjustable lens tube illuminators were used at the illumination arm and the detection arm. The illumination lens tube was coupled with an Ocean Optics DH-2000-BAL light source using a 50μm optical fibre to compromise between the spot size on the sample and its intensity. The lens tube on the detection arm was coupled with an Ocean Optics QE Pro spectrometer via a 400μm optical fibre to enhance signal-to-noise level. Both lens tubes were positioned at a distance of about 300 mm from the sample. The divergence of light collected by the detector at each angle position is about ±1°. The setup was carefully calibrated before measurements to ensure accuracy. A silver mirror was used as the reference for reflection measurements.

## Data availability

The authors declare that the main data supporting the findings of this study are available within the article and its Supplementary Information files. Data are available from https://doi.org/10.6084/m9.figshare.25931779. Extra data are available from the corresponding author upon request.

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

## Acknowledgements

The research is sponsored by the National Natural Science Foundation of China (51903155, 51802191, 52011530182) and the Shanghai Rising-Star Program (21QA1404700). Tan Sui and Qibin Zhao acknowledge the support from the Royal Society International Exchanges Cost Share (NSFC) scheme (IEC\NSFC\191003).

## Author contributions

Q.Z. and J.J.B. conceived the work, Q.Z. and T.A. designed the experiment, T.A., X.J., F.G., C.S., J.Q., X.S., M.Z., X. L., and F.T. performed the experiments. Q.Z. and T.A. analyzed the data. Q.Z, J.J.B., T.A., C.E.F., X.Z., B.Z., T.S., X.H., S. N., and T.F discussed the results. Q.Z. wrote the manuscript. All authors contributed to the review and editing of the manuscript.

## Competing interests

The authors declare no competing interests.
