## [Peer Review File · Nature Communications]

Strain to shine: stretching-induced three-dimensional symmetries in nanoparticle-assembled photonic crystalsREVIEWERS' COMMENTS:

Reviewer #1 (Remarks to the Author):

The manuscript deals with the interpretation of the optical response of artificial opals infiltrated with an elastomer (polymer opal, PO). Upon stretching the PO along different direction, an optical model accounting for different crystallographic direction as well as disorder is developed allowing for the interpretation of the data.

I don't think the manuscript suitable for publication in Nat. Comm. for 2 reasons.

1) Even if I agree with authors about the importance of understanding the microscopic mechanisms beyond the mechanochromic response of 3D elastomeric photonic crystals, the manuscript is very difficult to read and can be understood only by readers with a specific knowledge of the topics. It's not suitable for the broad readership of Nat. Comm.

2) The sentence at row 59 "3D photonic crystals, display only monotonous blueshift colours from the (111) planes parallel to the surface" is not true. And the following discussion uses a published model described in ref 49. There are several report published on red shift of optical modes in opal-like structures starting from early 2000.

I think authors have to better analyze their data might have an easier interpretation based on older literature. For instance, they use, ref 49 - published in 2009 - which is based on previous works one of that being there cited (PHYSICAL REVIEW B 78, 205304 2008) forgetting earlier reports on the topics (Phys. Rev. B 72, 045102 2005).

According to such papers, in order to understand the red shift observed, it is of paramount importance to focus the attention not only on the main pseudo gap but also on high energy structures in the spectra behaving as Van-Hove singularities - observed at photon energies above the main pseudogap. Van Hove singularities show a dispersion opposite to that of the main pseudogap. Indeed for incidence angle around 40-50 degrees the high energy red shifted band crosses the blue shifting main band gap.

Without a detailed comparison with the results of such old literature, the model here proposed does not show novelties, except for the role of disorder and possible cells rotations.

I think the manuscript - after a careful review for point 2) and perhaps some comment on Poisson ratio (as authors stretched the samples along two directions) could be suitable for publication in a more specialistic journal.

Reviewer #2 (Remarks to the Author):

Mechano-chromic photonic crystals (MPCs) with non-closely packed and inverse opals structures have attracted growing interest owing to their unique properties of regulating structural colors by stretching. Unfortunately, the evolution of MPCs' lattices and the variation of each plane during stretching are important but still unclear. This well-organized work systematically investigated the lattice transformation and optical change of three-dimensional MPCs with closely packed structures by stretching. Compared to non-closely packed structures, the evolution of the closely packed structure in this work is not attractive. In addition, similar results have already been reported by several groups (Chem. Mater., 2007,19, 23. ACS Applied Polymer Materials 2020, 2, 9, 4078-4089; Langmuir 2013, 29, 11275–11283; ACS Appl. Mater. Interfaces 2023, 15, 47350–47358; Chem. Mater. 2019, 31, 8918–8926; Small, 2020, 16, 1907626). The significance and novelty of this work does not meet the high standards of Nature Communication and it is more suitable to be published in a more specialized journal such as Advanced Optical Materials or ACS Applied Materials & Interfaces, etc.

Reviewer #3 (Remarks to the Author):

The paper is truly interesting and shows the vital connection between mechanics and optics, offering a broad avenue for new fundamental and applied research and applications.

I found no significant issues and fully agree with the authors' logic of presentation and their conclusion.

However, I have two minor remarks. First, authors should include in their reference list and mention one of the first studies of photonic crystals at a soft substrate that shows the effect of curvature and wrinkling(stress) on the optical response. (Kolaric & Damman et al, Design

of curved photonic crystal using swelling induced instabilities, J. Mater. Chem., 2012, 22, 16205-16208)

Second, in the conclusion, the link between optics and mechanics should be additionally highlighted, as well as potential applications based on that link.

In the end, I recommend a minor revision of this draft.

Reviewer #4 (Remarks to the Author):

This manuscript provides the account of a rather complete combined experimental and modelling work on stretching induced lattice transformations and the resultant optical properties.

This work appears to be the first such analysis and provides a framework for elaborating strain as a handle to control and modulate optical properties.

I am not in favour of SI and extended data, and even less so for the combination of both.

Although the authors provide an elaborate introduction and have compiled a long list of references, in my view, 2 crucial references where mechanical stretching for deformation, and in particular for colloidal crystals, are missing:

Jiang, Bertone and Colvin, Science 2001, 291, 453 "A lost-wax approach to monodisperse colloids and their crystals";

and

Tao Ding et al., Langmuir 2009, 25(17) 10218-10222 "Photonic crystals of oblate spheroids by blown film extrusion of prefabricated colloidal crystals".

Pending the consideration for including these references, my recommendation is to accept for publication in Nature Communications.

Responses to reviewers' comments:

Reviewer #1:

Comments:

The manuscript deals with the interpretation of the optical response of artificial opals infiltrated with an elastomer (polymer opal, PO). Upon stretching the PO along different direction, an optical model accounting for different crystallographic direction as well as disorder is developed allowing for the interpretation of the data.

I don't think the manuscript suitable for publication in Nat. Comm. for 2 reasons.

1) Even if I agree with authors about the importance of understanding the microscopic mechanisms beyond the mechanochromic response of 3D elastomeric photonic crystals, the manuscript is very difficult to read and can be understood only by readers with a specific knowledge of the topics. It's not suitable for the broad readership of Nat. Comm.

R: We are very grateful that the reviewer has raised concerns about the readability of the original manuscript. We fully agree that it is critically important for the manuscript to be presented in a manner that is more accessible to a wider audience across various disciplines.

In recognition of this issue, we have thoroughly revised the manuscript. While retaining the original data and focus, the entire introduction section is rewritten to improve organization, clarity, and provide a gradual introduction to the background, broader implications, and novelty of our work. Additionally, we have moved equations and discussions to the Supplementary Information (SI) and added new discussions and results. These additions, such as the symmetry breaking of the Brillouin zone, reveal a more explicit link between the evolving 3D lattice structures and the optical properties, further highlighting the novelties of our research. We also realized that the original figures were cluttered and difficult to interpret, leading us to redraw the figures as Figures 1-5 in the revised manuscript. Moreover, we have included an extra figure, Figure 6, which demonstrates the various optical effects that can be achieved through 3D lattice deformation and explicitly links to potential applications. These changes do not alter our original conclusions; rather, they better highlight our novelties by providing discussions and details from different perspectives. We sincerely hope the revised manuscript is now much easier to read and more appealing to a broader readership.

2) The sentence at row 59 "3D photonic crystals, display only monotonous blueshift colours from the (111) planes parallel to the surface" is not true. And the following discussion uses a published model described in ref 49. There are several report published on red shift of optical modes in opal-like structures starting from early 2000. I think authors have to better analyze their data might have an easier interpretation based on older literature. For instance, they use, ref 49 - published in 2009 - which is based on previous works one of that being there cited (PHYSICAL REVIEW B 78, 205304 2008) forgetting earleir reports on the topics (Phys. Rev. B 72, 045102 2005). According to such papers, in order to understand the red shift observed, it is of paramount importance to focus the attention not only on the main pseudo gap but also on high energy structures in the spectra behaving as Van-Hove singularituies - observed at photon energies above the main pseudogap. Van Hove singularities show a dispersion opposite to that of the main pseudogap. Indeed for incidence angle around 40-50 degrees the high energy red shifted band crossess the blue shifting main band gap. Without a detailed comparison with

the results of such old literature, the model here proposed does not show novelties, except for the role of disorder and possible cells rotations.

I think the manuscript - after a careful review for point 2) and perhaps some comment on Poisson ratio (as authors stretched the samples along two directions) could be suitable for publication in a more specialistic journal.

R: We greatly appreciate the reviewer's comments and fully understand why the concern was raised. In fact, we completely agree with the reviewer regarding the van Hove singularity related optical effects. We are also pleased that the reviewer highly values the optical effects from multiple different photonic structures in opals, not just limited to the main pseudogap of the (111) planes. This aligns with the aims and focus of our manuscript: to reveal how the 3D lattice changes and the evolution of optical properties in multiple photonic structures. We find that the impression of disagreement is largely due to the writing of the original manuscript, which, as mentioned by the reviewer, was difficult to read and has led to an apparent misunderstanding.

Due to the original manuscript's length limitations, the introduction was very concisely written, but perhaps it was oversimplified, leading to misunderstandings. We were aware that much previous research on synthetic opals has addressed optical phenomena while changing the incident angle (otherwise, we wouldn't have cited the PRB papers). Indeed, for synthetic opals with a fixed structure, increasing the angle of incidence from 0° in the normal direction initially causes a blueshift of the main stopband. Then, as the reviewer mentioned, at large angles, a different peak appears and shifts to longer wavelengths, opposite to that of the main pseudogap. From the late 1990s to the 2010s, there has been intense discussions on the mechanisms of this phenomenon, with some examples including:

1. Pavarini, E. *et al.* Band Structure and Optical Properties of Opal Photonic Crystals. *Phys Rev B* 72, (2005).
2. Galisteo-López J. F., *et al.* Optical study of the pseudogap in thickness and orientation controlled artificial opals. *Phys Rev B* 68, (2003)
3. Romanov, S. G. *et al.* Diffraction of Light from Thin-film Polymethylmethacrylate Opaline Photonic Crystals. *Phys Rev E* 63, 566031–566035 (2001).
4. van Driel, H. M. & Vos, W. L. Multiple Bragg Wave Coupling in Photonic Band-gap Crystals. *Phys Rev B* 62, 9872–9875 (2000).

There are many other publications on this topic, but the four listed papers are particularly representative and correlated. The detailed discussions in the work by Pavarini, E. *et al.*, as mentioned by the reviewer, introduce a mechanism based on van Hove singularities by calculating the reduced density of states at varying angles. In the paper, it explicitly attributes the new features to contributions from the (200) and other {111} planes that are tilted relative to the surface (111) planes. At the bottom of page 7, the authors compare their results with previous works, stating: “The present interpretation does not contradict the multiple Bragg-wave coupling or band branching mechanisms previously introduced, and it confirms the identification of multiple diffraction as arising from {111} or {200} family planes for the different orientations. Extending previous works, our observations indicate that diffraction-related features in reflectance spectra become increasingly visible for higher values of the angle of incidence: i.e., the onset of diffraction, or equivalently, the excitation of several photonic modes at the same frequency, is not restricted to the vicinity of the W or K/U points of the Brillouin zone and should be viewed as a more general characteristic of the optical response of the investigated photonic crystals.”

Aware of the exemplary work on synthetic opals, we specified in line 59 of the original manuscript, before '3D photonic crystals display only monotonous blueshift colours from the

(111) planes parallel to the surface', by prefacing with 'For instance, when stretched, elastic opals, known as 3D photonic crystals, display only monotonous blueshift colours from the (111) planes parallel to the surface.' Our intention was to limit the discussion to the stretch-induced optical properties of elastic opals. This focus was chosen because previous works addressing the stretch-induced colour changes of elastic opals predominantly concentrated on the optical shifts of the main (111) pseudogap. This is precisely why we undertook our work: to reveal strain-induced changes of the optical properties from other lattice structures. However, it appears the original phrasing was too subtle to address the issue clearly and might have led to misunderstandings.

In the revised manuscript, we have significantly improved the introduction, especially in the third and fourth paragraphs, where we detail issues related to the reviewer's concerns and added the listed references. Furthermore, in the section 'Optical Transitions in n -Stretching,' we elaborate on our approach with discussions on previous work from the references listed above added more discussions related to the evolutions of the 1st Brillouin zone. We demonstrate how the main pseudogap (111) planes move out of the 1st Brillouin zone with stretching, and how the optical properties of the stretched material become increasingly dependent on the tilted planes instead of (111). Their dynamic evolutions involve the merging and appearing of different symmetry points in the Brillouin zone and the related possible optical singularities. We hope these changes enhance the manuscript's clarity, and we wish to thank the reviewer again for raising this issue.

Reviewer #2:

Comments:

Mechano-chromic photonic crystals (MPCs) with non-closely packed and inverse opals structures have attracted growing interest owing to their unique properties of regulating structural colors by stretching. Unfortunately, the evolution of MPCs' lattices and the variation of each plane during stretching are important but still unclear.

R: We appreciate the reviewer's comments. Indeed, as mentioned by the reviewer, the evolution of mechano-chromic photonic crystals (MPCs) with non-closely packed opal's lattices and the variation of each plane during stretching are important but still unclear. This aligns with the aims and focus of our manuscript: to reveal how the 3D lattice changes and the evolution of optical properties in multiple photonic structures.

This well-organized work systematically investigated the lattice transformation and optical change of three-dimensional MPCs with closely packed structures by stretching. Compared to non-closely packed structures, the evolution of the closely packed structure in this work is not attractive.

R: We are very grateful about the reviewer's appreciation of our work. We agree that the evolution of non-closely packed structures is truly interesting.

We assume that the non-closely packed structures mentioned by the reviewer refer to lattice structures other than *fcc*, *hcp* or *r-hcp*; thus, these structures include *bcc*, triclinic, monoclinic, and simple cubic, *etc.* As mentioned in our manuscript, we use stretchable 3D photonic crystals with a close-packing lattice as a prototype to study the 3D lattice changes and optical transitions of multiple photonic structures, and our findings offer a viable approach that can be applied to materials with various lattice structures. In fact, the deformation of a typical non-closely packed lattice structure, such as the triclinic lattice, is already included in our

manuscript: when the material is stretched in n -direction, its structure immediately turns into a triclinic lattice, and further stretching induces continuous deformation of the triclinic lattice. The related optical transitions in the triclinic lattice deformation are also introduced in detail in the section “Optical transitions in n -stretching”. In fact, p -stretching also involves the deformation of non-closely packed lattices. Note that the intermediate states from 0% strain to ~40% strain and those above ~40% strain are all non-closely packed lattices. Thus, although we start with a close-packed lattice, the entire process encompasses the deformation of various lattice structures, all of which are predicted by our model. The reduced readability from the phrasing and organization of the original manuscript may have obscured these features. Therefore, we have made significant improvements to both the main text and the figures.

To avoid confusion in terms, in the introduction of the revised manuscript, we particularly introduced the difference between non-closely packed responsive photonic crystal and non-closely packed lattice structures. In the field of responsive 3D photonic crystals, the term “non-closely packed responsive 3D photonic crystals” is often used to indicate that spheres are not directly in contact, compared to synthetic opals where spheres touch closely. The term has little to do with the lattice structure, and most of the “non-closely packed responsive 3D photonic crystals” developed so far are of a close-packing lattice. In the early days, in the field of colloidal science a different term “loosely packed” is often used instead of non-closely packed. However, the new term become popular in recent decades. We include some references on 'non-closely packed responsive 3D photonic crystals' here, as the reviewer may already be aware of them:

1. Ge, J. & Yin, Y. Responsive Photonic Crystals. *Angewandte Chemie International Edition* 50, 1492–1522 (2011).
2. Hu, Y., *et al.* Stimulus-responsive non-close-packed photonic crystals: fabrications and applications”, *Mater. Hori.* 10, 3895 (2023)

As we have also mentioned in the introduction of the revised manuscript, non-close-packed 3D photonic crystals of lattice structure other than fcc , hcp or r - hcp , are possible. Examples of such materials are shown in the works by Yadong Yin’s group (Coupling morphological and magnetic anisotropy for assembling tetragonal colloidal crystals, *Science Advances*, 2021). However, most of the materials developed so far still rely on the close-packing lattices. According to the definition, the material we use in the manuscript is a typical non-closely packed 3D photonic crystal since in the first paragraph of the discussions section we have clearly stated that the spheres are not directly in contact and the soft matrix takes 55% volume ratio in the material. Nonetheless, despite discussions on the terminology, in the revised manuscript, we provide a clear pathway for studying the deformation of various lattice structures and reveal the fundamental principles governing their deformation. These challenges can be analytically addressed with boundary conditions set by the fundamental principles and the specific geometric relations in each lattice type. The deformations of some typical non-closely packed lattices, such as triclinic lattices, have already been demonstrated in our model. We sincerely hope the revised version can adequately address the reviewer’s concerns.

In addition, similar results have already been reported by several groups (*Chem. Mater.*, 2007,19, 23. *ACS Applied Polymer Materials* 2020, 2, 9, 4078-4089; *Langmuir* 2013, 29, 11275–11283; *ACS Appl. Mater. Interfaces* 2023, 15, 47350–47358; *Chem. Mater.* 2019, 31, 8918–8926; *Small*, 2020, 16, 1907626). The significance and novelty of this work does not meet the high standards of *Nature Communication* and it is more suitable to be published in a more specialized journal such as *Advanced Optical Materials* or *ACS Applied Materials & Interfaces*, etc.

R: We appreciate the reviewer's comments and the discussions based on the referenced works. We agree that it is widely accepted that stretching will induce 3D lattice changes in materials. However, as the reviewer has mentioned in earlier comments, how the 3D lattice change and the mechanism behind these changes is not well understood. Achieving a better understanding always requires gradual progress, built upon the collective efforts of many researchers over many years.

The first two references mentioned by the reviewer, 'Chem. Mater., 2007, 19, 23' and 'ACS Applied Polymer Materials 2020, 2, 9, 4078-4089', were included in the original manuscript's references. We are pleased that the reviewer also finds these excellent works very interesting. The third reference, 'Langmuir 2013, 29, 11275–11283', comes from our collaborator, with its first author being one of our co-authors. Due to the limited number of references allowed in our manuscript, it was not included in our list. Indeed, the reference 'Chem. Mater., 2007, 19, 23' also originates from our collaborators at DKI/TU. Darmstadt. We are well acquainted with the work, having collaborated on the polymer opal project for the last 20 years. A selection of our collaborative papers includes:

1. Compact strain-sensitive flexible photonic crystals for sensors. *Appl. Phys. Lett.* 87, 101902 (2005)
2. 3D Bulk Ordering in Macroscopic Solid Opaline Films by Edge-Induced Rotational Shearing. *Adv. Mater.* 23, 1540 (2011)
3. Electric-field-tuned color in photonic crystal elastomers. *Appl. Phys. Lett.* 100, 101902 (2012)
4. Large-scale ordering of nanoparticles using viscoelastic shear processing. *Nat Commun* 7, 11661 (2016)

Over the years, we have been working together to build knowledge in a step-by-step manner toward a complete understanding of 3D lattice transformations and to fully reveal the rich optical evolutions. We have recognized the limitations in previous results, which motivated the work presented in our manuscript. The references 'Chem. Mater. 2019, 31, 8918–8926,' and 'Small, 2020, 16, 1907626' offer refreshing ideas and results on the fabrication and applications of novel stretchable 3D photonic crystals. Unfortunately, the change in various lattice structures and the optical changes of multiple photonic bandgaps are not the main focus of these works, as they primarily concentrate on the shifting of the main stopband of (111) planes. The 'ACS Appl. Mater. Interfaces 2023, 15, 47350–47358' paper reports the retro-reflecting colour found in stretched opals, a phenomenon also observed in our study. In fact, we discovered this phenomenon in polymer opals many years ago (evidence available in data archives, research proposals, and the PhD thesis 'Optomechanical Anisotropy in Nanoengineered Polymer Photonic Crystals' by Kontogeorgos), but we were never able to fully understand its structural origin. This curiosity partly fuelled our continuous work over the past years. In the referenced paper, the authors attribute it to the diffraction within the (111) planes, akin to the diffraction of a 2D monolayer, thus not linking it to other lattice planes of the 3D lattice. We offer a different explanation, which arises from the photonic bandgaps of tilted planes such as $(\bar{1}11)$ and (200) planes. Despite our disagreement on the mechanism, we are pleased to see that such a phenomenon has been noticed and valued by others, and we applaud their findings.

The references mentioned by the reviewer all exhibit significant novelties within their respective areas of focus, but they predominantly concentrate on the (111) planes. Our work, in contrast, explores the changing 3D symmetry of the lattice, encompassing various lattice planes, and provides an explicit link between 3D lattice deformation and the evolution of the Brillouin zone in k -space. Our results demonstrate how the main pseudogap (111) Bragg planes move out of the 1st Brillouin zone upon stretching, and how the optical properties of the stretched material become increasingly dependent on the photonic bandgaps of tilted planes,

rather than just on (111). The evolutions involving the merging and appearance of different symmetry points open up promising avenues for further studies in topological photonic crystals. In the revised manuscript, we have also included a new subsection and Fig. 6 to specifically address potential applications related to tuning tilted planes through strain. To our knowledge, the detailed stretching-induced 3D lattice deformations and the evolutions of the complex interplay of multiple photonic bandgaps are unprecedented.

We truly appreciate the contributions of various groups to this field; such collective efforts are the cornerstone of scientific advancement. We sincerely hope that our work will further contribute to the development of these materials.

Reviewer #3:

Comments:

The paper is truly interesting and shows the vital connection between mechanics and optics, offering a broad avenue for new fundamental and applied research and applications. I found no significant issues and fully agree with the authors' logic of presentation and their conclusion.

However, I have two minor remarks. First, authors should include in their reference list and mention one of the first studies of photonic crystals at a soft substrate that shows the effect of curvature and wrinkling(stress) on the optical response. (Kolaric & Damman et al, Design of curved photonic crystal using swelling induced instabilities, *J. Mater. Chem.*, 2012, 22, 16205-16208).

R: We greatly appreciate the reviewer's advice. Indeed, for decades, people have been fascinated by the interesting optical properties of colloidal particles assembled into photonic crystals, and extensive research has been conducted. The work mentioned by the reviewer utilizes mechanical instability-induced curved structures on the surface of a soft polymer substrate for the assembly of colloidal photonic crystals. This enables the resultant material to display structural colours that are nearly angle-independent. This feature is particularly noteworthy because most 3D photonic crystals are fabricated on flat surfaces and display angle-dependent colours.

The conventional approach to achieving angle-independent colours involves using photonic glass materials, but their reflection intensity is typically low compared to that of Bragg reflection. The approach introduced by Kolaric & Damman *et al.* maintains the high-intensity Bragg reflection from colloidal crystals and redistributes its direction across various angles using microscale curvatures, a method that has proven to be facile in fabrication. We have added this work as reference [2] in our manuscript.

Second, in the conclusion, the link between optics and mechanics should be additionally highlighted, as well as potential applications based on that link.

In the end, I recommend a minor revision of this draft.

R: We are very grateful for the reviewer's advice. We have rewritten the introduction section of our manuscript to provide a more detailed background for this work. In response to the reviewer's suggestions, we have also revised the results section, especially the subsection titled "Optical Transitions in *n*-Stretching". We have added more discussions on the optical phenomena induced by mechanical structure changes and have illustrated these more clearly in the revised figures. Additionally, we have included a new Figure 6 and its accompanying

discussion in the subsection “Light Tuning with Strain” to demonstrate various optical effects that can be used for diverse applications.

Reviewer #4:

Comments:

This manuscript provides the account of a rather complete combined experimental and modelling work on stretching induced lattice transformations and the resultant optical properties.

This work appears to be the first such analysis and provides a framework for elaborating strain as a handle to control and modulate optical properties.

I am not in favour of SI and extended data, and even less so for the combination of both.

Although the authors provide an elaborate introduction and have compiled a long list of references, in my view, 2 crucial references where mechanical stretching for deformation, and in particular for colloidal crystals, are missing:

Jiang, Bertone and Colvin, Science 2001, 291, 453 "A lost-wax approach to monodisperse colloids and their crystals";

and

Tao Ding et al., Langmuir 2009, 25(17) 10218-10222 "Photonic crystals of oblate spheroids by blown film extrusion of prefabricated colloidal crystals".

R : We greatly appreciate the reviewer's comments. We agree that the use of SI (Supplementary Information) and extended data may have negatively affected readability. In light of this, we have revised the SI by reorganizing the figures and removing less important information.

We have added the work by Jiang, Bertone, and Colvin as reference [1], and the work by Tao Ding et al. as reference [3]. We have included these references in both the first and second paragraphs of the revised introduction. These works demonstrate how particles of different shapes can be achieved through deformation, highlighting the various effects attainable with mechanical deformation. Typically, nanoparticles of different shapes are obtained by controlling their chemical synthesis process. However, these works show how this can be realized in a more facile manner. Additionally, they reveal that mechanical deformation may not only change the spatial arrangement of the particles but also modify their shapes. In our work, we use highly crosslinked polystyrene spheres that remain spherical during stretching since the stress is low. However, if we use non-crosslinked polystyrene spheres and apply high stress, their shape may also change, as shown in the works mentioned by the reviewer. It would be very interesting to see how the lattice structures of such particles change. We would like to thank the reviewer again for providing references that offer new inspirations for future work.

REVIEWERS' COMMENTS:

Reviewer #1 (Remarks to the Author):

I appreciated the author's effort to improve the paper. However, I still find it unsuitable to the broad readership of Nat. Comm.

I don't find there significant novelties to meet the goals of Nature Communication and to deeply impact the community working in the field.

However, the manuscript is for sure solid and deserve to be published in a more specialistic journal.

Reviewer #2 (Remarks to the Author):

The authors have partially addressed my questions in their revised manuscript. Although I believe that retroreflection color arises from the photonic bandgaps of tilted planes such as (T11) and (200) planes; however, I cannot be fully convinced. This is because the conclusion is significantly different from the previous works (Chem. Mater. 2019, 31, 8918–8926 and ACS Appl. Mater. Interfaces 2023, 15, 47350–47358) that both suggest that retroreflection comes from the 2D Bragg diffraction under high strain. I am still confused whether the retroreflective color comes from 2D diffraction or tilted planes such as (T11) and (200) planes. I am confused since both mechanisms seem reasonable. Therefore, I cannot recommend the acceptance of this work unless the authors have addressed the following issues.

1. It is suggested that the authors provide the surface SEM images of the PCs with different strains along both n- and p-direction, which are important to understand whether 2D Bragg diffraction contributes to the retroreflection color.
2. Will the retroreflection wavelength and color be different between n- and p-directions?
3. How does reflectance change during stretching? The real reflection spectra of the PC under different strains should be supplied.
4. Is it possible to compare the color saturation between the reflection and retroreflection?
5. What are the possible applications of the elastic PCs with two different force-responsive colors?

Responses to reviewers' comments:

Reviewer #1:

Comments:

I appreciated the author's effort to improve the paper. However, I still find it unsuitable to the broad readership of Nat. Comm. I don't find there significant novelties to meet the goals of Nature Communication and to deeply impact the community working in the field.

However, the manuscript is for sure solid and deserve to be published in a more specialistic journal.

R: We really appreciate the reviewer's comments, particularly their consideration of our work as solid. We have provided new experimental results using various colloidal-assembled elastic opals, distinct from the polymer opals (POs) used in our study, as depicted in Figs. S20-S24. These elastic opals are frequently featured in research on responsive 3D photonic crystals; however, previous studies have primarily focused on documenting their specular reflection colours. Our experiments demonstrate that these materials can exhibit colours in both specular and retro-reflection directions when stretched. Consistent with our findings using POs, the retro-reflection colours can consist of diffractions from surface structures as well as strong retro-Bragg reflections from higher-energy photonic bandgap structures, corroborating our theory. Notably, the retro-Bragg reflections of tilted planes have not been previously demonstrated in these materials. The results reveal that elastic opals can display distinct Bragg reflection colours in reflection when viewed from forward and retro-directions, which promises much wider applications. We hope this addition will broaden the appeal to a wider readership and enhance the impact on the community.

Reviewer #2:

Comments:

The authors have partially addressed my questions in their revised manuscript. Although I believe that retroreflection color arises from the photonic bandgaps of tilted planes such as $(\bar{1}11)$ and (200) planes; however, I cannot be fully convinced. This is because the conclusion is significantly different from the previous works (Chem. Mater. 2019, 31, 8918–8926 and ACS Appl. Mater. Interfaces 2023, 15, 47350–47358) that both suggest that retroreflection comes from the 2D Bragg diffraction under high strain. I am still confused whether the retroreflective color comes from 2D diffraction or tilted planes such as $(\bar{1}11)$ and (200) planes. I am confused since both mechanisms seem reasonable. Therefore, I cannot recommend the acceptance of this work unless the authors have addressed the following issues.

R: We appreciate the reviewer's comments and fully understand their concerns. Indeed, the reviewer has raised a very interesting question regarding the roles of 2D diffraction and retro-Bragg reflection from the $(\bar{1}11)$ and (200) planes in generating retro-reflection colours, and their interrelationships. This aspect is a crucial part of our work and may also be perplexing to others since this phenomenon and its mechanisms have not been previously reported in opals. Strictly speaking, we should refer to 1D diffractions in the two distinct directions of the 2D

structure because when considering diffraction in either the n-direction or p-direction, we treat the 2D structure as two 1D diffraction gratings, each with a different period. However, using the term '2D diffraction' is more convenient in our discussion, as it specifically highlights the diffraction effects arising from the 2D structure, in contrast to the retro-Bragg reflections from tilted planes.

The Point of Contention

The reviewer's primary uncertainty appears to be: on one hand, 2D diffraction from surface structures is cited as producing retro-reflection colours in the two referenced papers, and on the other, our work attributes retro-reflection colours to Bragg reflections from the $(\bar{1}11)$ and (200) planes. So, which interpretation is correct? In fact, both explanations can be correct. These two mechanisms are not mutually exclusive but coexist, and they have been both predicted in our theory. It might seem, based on the reviewer's references, that there is a conflicting origin of the retro-reflection colours compared to our findings, which could lead to confusion. However, upon closer examination of the specifics, such as strain levels, angles, and sphere sizes, there are no contradictions. The original wording in the first paragraph of the 'Optical transitions in *n*-stretching' section might have been slightly misleading, possibly leading readers to believe we disagree with the findings in the referenced paper, such as the AMI paper. This was not our intention, and we have since revised that paragraph to eliminate any potential misunderstandings. Here we clarify why the 2D diffraction mentioned by the reviewer does not contradict our theoretical framework.

Interplay of 2D Diffraction and Retro-Bragg Reflections

First, as we've mentioned in our manuscript, particularly in the last paragraph of the 'Optical transitions in *n*-stretching' section, we observe both 2D diffraction colors and retro-Bragg reflections of the $(\bar{1}11)$ and (200) planes in our samples. It's crucial to note that 2D diffraction is inherently present in opals, whether stretched or not. Stretching primarily enhances the visibility of 2D diffraction in the visible wavelength range and tunes its visible angle when the original grating period is too small. For instance, using larger spheres (300nm-1 μ m) can render 2D diffraction colours visible even without stretching, though they may not appear in the retro-reflection direction.

Retro-Bragg reflections of titled planes, such as $(\bar{1}11)$ and (200), only become evident above a certain strain level because the planes must be tilted by stretching to specific angles to overcome total internal reflection and exit the surface. This effect is irrespective of sphere size and relates to the tilting angle of the planes. Essentially, if you have an elastic opal, regardless of sphere size, when you fix the angle of incident light and analyze the scattering light at different angles, you will always detect signals of 2D diffraction. The challenge is whether its wavelength falls within your detector's range or at a desired direction. However, you will not detect retro-Bragg reflections of titled planes unless the sample is stretched. Furthermore, when you do detect retro-Bragg reflections at certain strains, 2D diffraction continues to exist. At this juncture, the relationship between 2D diffraction and retro-Bragg reflections of titled planes is particularly intriguing because the retro-Bragg reflections precisely overlap with the 2D diffraction light of the same wavelength and direction.

To illustrate their relationship, imagine a lighting fixture on a ceiling casting a rainbow pattern on the floor, where suddenly one specific colour in the pattern is significantly enhanced due to the addition of a laser shooting light of that colour in the same direction. Here, the rainbow

pattern represents 2D diffraction, and the strong laser light represents the retro-Bragg reflection of titled planes. The rigorous mathematical relations and theory are detailed in SI Discussion 2. This is fascinating as it reveals the fundamental connection between the optical performance of different structures in the 3D lattice.

According to model calculations of the retro-Bragg reflections of $(\bar{1}11)$ and (200) planes in n -stretched elastic opals of an ideal fcc lattice, shown in Fig. S12c, at 40% strain, one can only expect very weak retro-Bragg reflection from (200) planes above a viewing angle of 80° in the retro-reflection direction, provided the incident light is nearly parallel to the surface of the sample. Practically, this colour is barely visible if the material possesses an fcc structure. Weak retro-Bragg reflection signals in 40% n -stretched POs are observable due to the r -hcp structure's large number of stacking faults affecting the orientations of titled planes, as discussed in our manuscript. At 50% strain, more titled planes contribute to retro-Bragg reflection, yet these colours remain barely visible above 60° unless the incident light is at very high angles, and retro-Bragg reflections from $(\bar{1}11)$ planes are still not visible at any angle. When stretching reaches 60%, retro-Bragg reflections become significant above 60° because the $(\bar{1}11)$ planes start to contribute significantly. As stretching increases, the retro-Bragg reflections become visible at progressively lower angles. We did not depict the 2D diffraction spectra in Fig. S12c as varying the incident angle would complicate the plot, but we have illustrated expected 2D diffraction spectra at a 50° incidence for 60% and 80% n -stretched samples in Fig. 4e, marked with green lines.

In contrast to what's shown in the referenced AMI paper, the vertical height contrast in our POs' 2D surface grating is significantly less, as seen in Fig. S19. Before stretching, only the tiny tops of the spheres are visible, and even after 80% n -stretching, the vertical displacements of the spheres remain subtle. This starkly contrasts with the elastic opals used in the AMI paper, where the separation between spheres and the matrix material during fabrication results in a far more pronounced vertical height contrast, leading to stronger 2D diffraction intensities.

Comparative Analysis with the AMI and CM Papers' Findings

Let's now compare the results presented in the AMI paper with our theory, focusing on deformation-induced colour changes, as the AMI paper provides comprehensive experimental characterization. It uses a red elastic opal, producing a $\sim 615\text{nm}$ reflection peak wavelength at normal incidence. When the sample is stretched to a maximum of 50% strain, retro-reflection rainbow colours are observed at angles below 60° . The authors attribute the red to blue colour transition to 2D diffractions, a conclusion we fully support. Although the initial stretching direction is in the p -direction, their deformed surface structure seems to undergo an n -stretching process after 30% strain, as shown in their supporting information images. This phenomenon could be due to line defects and different crystal domains within the (111) in-plane structure, and there can also have many other reasons; similar features are observed in POs, as illustrated in Fig. S17e. However, this is not very important. Regardless of whether it is p -stretching or n -stretching, the crucial aspect of their work is the final structure at $\sim 50\%$ strain and its relationship to the retro-reflection colours. According to our theory and experimental results, no retro-Bragg reflections from tilted planes are observable at 50% strain whether it is n - or p -stretching. Details related to p -stretching-induced lattice transformations and optical transitions are provided in the ' p -stretching' section of our manuscript. For n -stretching, our model calculations, discussed earlier, indicate that retro-Bragg reflections from tilted planes are unlikely to appear at 50% strain below a viewing angle of 60° ; thus, the observed red to blue retro-reflection colours in their study are solely due to 2D diffraction, confirming their

conclusions. If the authors had stretched their samples slightly more, to 60% strain, retro-Bragg reflection colours from tilted planes would be observable at higher angles. The visibility of red to blue 2D diffraction colours at 50% strain in their stretched samples is attributable to the use of red opals, which produce a larger surface grating period.

While retro-reflection colour is not a primary focus of the CM paper and lacks detailed information on stretching directions and structures, it is challenging to discuss further. However, we concur with the 2D diffraction mechanism, especially since at 150% strain, the structural order significantly deteriorates. As depicted in Fig. S14 of our manuscript, beyond 120% strain, bright spots indicating strong retro-Bragg reflection from tilted planes disappear due to poor structural order, leading 2D surface diffraction to likely dominate the colour, as stretching-induced disorder typically has a more pronounced effect on 3D structures than on the 2D surface layer.

Experimental Validation Using Elastic Opals Similar to Those in Referenced Studies

To further justify our conclusions and provide a clearer answer to the reviewer's concerns, we present new experimental results using elastic opals composed of polystyrene (PS) spheres and an acrylate matrix. Unlike in previous experiments where the materials were *r*-hcp, here the PS spheres and acrylates are separated before assembly and subsequently form a highly ordered *fcc* lattice by shearing. This setup reduces concerns about structural disorder, closely aligning the materials with our theoretical model predictions. These new elastic opals are similar to those used in the AMI paper, except that we use PS spheres instead of silica. The results are presented in Figs. S20-S24.

We fabricated samples using spheres of various sizes. The red sample, made with slightly larger spheres, produces a reflection peak at normal incidence of ~650nm, while the green sample, made with slightly smaller spheres, produces a peak at ~550nm. These choices were deliberate to closely match the materials used in the AMI paper and our manuscript: the 650nm red opal closely resembles the 615nm red opal, and the 550nm green opal mirrors the 560nm PO used in our studies.

Fig. S20 shows the colour changes and surface structure deformation of the red opal during n- and p-stretching at different strains, aligning with our lattice deformation theory. Most critically, Figs. S21-S24 image the retro-reflection colours at various strains and measure the angular spectroscopy. For the red elastic opal, Fig. S21 illustrates a rainbow band from low strains due to 2D diffraction and a particularly strong colour starting above 60% strain from the retro-Bragg reflection of tilted planes. This pattern is more distinctly displayed in the spectra in Fig. S22, where at low strains and large angles, a clear arc or line evolves, representing the spectrum of 2D diffraction. Above 60% strain, a strong spot overlays the 2D diffraction arc, indicating the retro-Bragg reflection of tilted planes. These spectral characterization results perfectly align with our model predictions and confirm the conclusions of the AMI paper—that below 60% strain, only 2D diffraction is observable.

Fig. S23 shows that with stretching, both the top and bottom surface structures of the sample deform, resulting in first-order diffraction in both reflection and transmission, though retro-Bragg reflections of tilted planes do not appear on the transmission side.

The findings for the green opals are consistent; due to the smaller period of the 2D grating, the spectra in Fig. S24 do not show a rainbow band at low strains as seen with the red opals.

However, above 60% strain, the retro-Bragg reflections become pronounced and merge with the colours from 2D diffraction. Notably, above 120% strain, as the incident angle increases, the energy of the retro-Bragg reflection disperses more evenly across the arc of the 2D diffraction spectra, a phenomenon related to the excessive tilting of the planes during stretching.

Given that 2D diffraction is too weak to be characterized spectroscopically in our POs due to the use of core-shell spheres and poorer structural order, there may be skepticism about the optical mechanisms. By using the new elastic opals, we hope to dispel any doubts. The relationship between the retro-Bragg reflection of tilted planes and 2D diffraction is clearly demonstrated in Fig. S22b.

We trust that this detailed explanation and the experimental results will help clarify any confusion the reviewer may have had.

1. It is suggested that the authors provide the surface SEM images of the PCs with different strains along both n- and p-direction, which are important to understand whether 2D Bragg diffraction contributes to the retroreflection color.

R: We are very grateful for the reviewer's suggestion. We've provided SEM and AFM images of the surface structures of the original and *n*-stretched POs in Fig. S19. Clearly, a surface grating structure exists, but when compared to the SEM images of the newly made elastic opals shown in Fig. S20, you may notice a significant difference. The surface SEM images of the new elastic opals resemble those in the AMI paper, where individual spheres are clearly visible, looking nice and clean. We absolutely agree that surface SEM images would be the most straightforward way to reveal the 2D structure changes at the surface. The main reason we have chosen to develop more sophisticated alternative characterization methods over SEM to reveal the structures is the really poor imaging quality of SEM for POs, which is hardly solvable.

Since we began working on polymer opals twenty years ago, we have spent tremendous effort trying to characterize the structures using SEM, but it has proven to be extremely difficult. We have tried different methods, including etching the surface with solvents and plasma etching, but they have not proved very useful. This is an intrinsic drawback of using core-shell spheres. The shell material is chemically grown on the core spheres during synthesis; after assembly, as seen in Fig. S19a, the spheres are deeply embedded under the surface, and even after stretching, the matrix material is still strongly bonded to the spheres at the surface, which gives limited protrusion of the spheres and no clear gap between them, resulting in very poor conductivity and low contrast for imaging. The difference in imaging the surface structures with SEM for elastic opals that are made of separate spheres and matrix materials, and POs using core-shell spheres, is akin to comparing rice in water before and after boiling into porridge. Considering the properties of the materials used in CM paper, we assume they may have similar problems with SEM imaging.

The 2D surface structure deformation is the deformation of the in-plane structure of the (111) plane, which can be characterized by alternative solutions. In our manuscript, we've provided results with two different methods, as shown in Figs. 3 and 5. The first method uses TEM. To image the in-plane structure of the (111) planes, we cut very thin slices parallel to the surface. The images of the (111) planes were obtained using slices cut at a distance of approximately 500 nm to 1 μ m very next to the surface, which should accurately represent the 2D structure changes. To double-check the obtained structure, we also performed SAXS measurements normal to the surface. Because the diameter of the SAXS beam is about 300 μ m, the diffraction

pattern can reveal the (111) in-plane structure over a large area. We also shifted the focus of the beam to different areas over centimetre scales of the sample to ensure that there is no orientation difference. The structure information obtained using different methods allows us to consider the 2D diffraction effects in stretching without relying on SEM images. In fact, as provided in previous discussions, 2D diffraction has already been included in our mechanism, and in our manuscript, we've also mentioned that we observe both the 2D diffraction and retro-Bragg reflection simultaneously, with their relationships described in detail. We hope this may address the reviewer's concern.

Additionally, we've provided SEM images using new elastic opals and their optical characterization results, which confirm our conclusions. Detailed discussions are provided before this question, and we really hope this will help to provide a clear understanding of the roles of 2D diffraction and retro-Bragg reflection of tilted planes.

2. Will the retroreflection wavelength and color be different between *n*- and *p*-directions?

R : Yes, they are completely different. The scattering colours and spectroscopy characterizations for *p*-stretching are shown in Fig. S17. The reason we didn't include Fig. S17 in the main text of the manuscript is that we are reaching the length limit, and it would require quite a lot of words to explain all the features in detail, which does not add significant new value to our work. To better contrast their colour differences, the angular scattering colour imaging of *n*- and *p*-stretching is provided in Fig. S18. As the reviewer is aware, there are two mechanisms for retro-reflection colour in stretching: 2D diffraction and retro-Bragg reflection of tilted planes. Regarding 2D diffraction, the grating period differs between the *n*- and *p*-directions; thus, even with the same amount of strain, you will always observe distinct 2D diffraction spectra in these two directions. The period differences in the *n*- and *p*-directions are also clearly presented in this reference (Ohnuki R, et al., *Advanced Optical Materials*, 2019, 1900227, 7, 13). The exact 2D diffraction spectra can be calculated using the grating period and incident angle, which provide the distribution of different wavelengths of diffracted light at various angles. However, as mentioned in the manuscript, the period of the grating in the *p*-direction after 160% *p*-stretching becomes equal to the period after 40% *n*-stretching in the *n*-direction. Thus, the wavelength of the 2D diffraction colours after 160% *p*-stretching becomes similar to those after 40% *n*-stretching.

Regarding the retro-Bragg reflection wavelength, it entirely depends on the plane distance and tilting angle of the tilted planes. If you consider the *fcc* lattice structure, different lattice planes have different orientations; for example, ($\bar{1}11$) and (200) planes only tilt in the *n*-direction. Due to the symmetry of the *fcc* lattice, the planes that are tilted in the *p*-direction only have very high indices, which means their lattice distances are much smaller than those of (111), ($\bar{1}11$), and (200) planes, making it difficult to observe their retro-Bragg reflections. As introduced in the manuscript and shown in Fig. S17, retro-Bragg reflections in *p*-stretching may only be observable above 160% strain in green opals.

The wavelengths of the retro-reflection light and its angular distribution always depend on the strain and incident angle, so when comparing the colours in *n*- and *p*-stretching, we must always consider specific conditions.

3. How does reflectance change during stretching? The real reflection spectra of the PC under different strains should be supplied.

R: We appreciate the reviewer's comments. The reflection spectra at normal incidence during stretching are provided in Fig. S3. At normal incidence, the reflectance of the (111) Bragg reflection peak consistently decreases, which aligns with many previous studies. It also reveals the reflection peak of tilted planes in n-stretching. However, it is important to note that the Bragg reflection signal from the tilted planes here is not the retro-Bragg reflection of the tilted planes; it is part of the specular reflection, as mentioned in our manuscript. This reflection involves the von Hove singularity. We have also included reflectance information on the colour scale bar alongside the angular spectroscopy in the supplementary information figures.

4. Is it possible to compare the color saturation between the reflection and retroreflection?

R: This is indeed a very interesting question since colour saturation is often a significant concern in real applications. Colour can be evaluated in different ways, such as through RGB, CMYK, or HSB formats, where 'H' stands for hue, 'S' for saturation, and 'B' for brightness. Saturation does not necessarily depend solely on brightness or wavelength; it also concerns how pure or vivid the colour is. Saturation can be evaluated either subjectively or objectively. Subjectively, based on our experience, both retro-reflection and specular reflection colours are visually appealing. In an industrial setting, there are professional tools that calibrate colour saturation directly from measured spectra, though this is beyond our current capability. Another common method to judge saturation involves using software such as Photoshop to show the saturation value of selected pixels. For example, we have demonstrated the saturation values of the specular and retro-reflection colours of original and stretched POs in Fig. S25. These values were obtained using the HSB mode in image analysis, which directly shows the saturation value of specific pixels. As shown in the results, the saturation levels of the colours are similar, and colour saturation also depends on how the images are taken and from which angle the colours are viewed. High saturation does not necessarily imply high reflection quality or more intense colour. As seen in Fig. S25, the blue-shifted colour after stretching appears to have a higher saturation value compared to the original green colour before stretching; however, recalling their spectra in Fig. S3, where the intensity of reflection decreases with strain, it is difficult to determine which colour is superior based solely on saturation value. Thus, while it is indeed possible to compare colour saturation between reflection and retro-reflection colours, doing so effectively requires additional information to accurately evaluate the colours.

5. What are the possible applications of the elastic PCs with two different force-responsive colors?

R: In terms of 'two different force-responsive' colours, we interpret this to mean the specular reflection colours and the retro-reflection colours. Retro-reflection colours could be 2D diffraction colours, retro-Bragg reflection colours, or a mixture of both. In the manuscript, we introduced some potential applications such as waveguides and emission control. There are many other applications, for example, anti-counterfeiting. Opals are often considered promising candidate materials for anti-counterfeiting; however, they typically utilize only the Bragg reflection of the (111) planes, which provides only specular reflection colours. Elastic opals that exhibit two different force-responsive colours offer distinct visual effects between the forward and retro-reflection directions, enhancing their competitiveness in real applications. These materials can also act as switches in optical communications or computing. For instance, in optical circuits, they could allow light of different wavelengths to travel in different directions—green light traveling forward and red light reflecting back. Using these materials, it is possible to efficiently separate a single incident beam into multiple wavelengths in different directions, with this modulation being flexibly tuned by applying or releasing strain

with an external stimulus. The practical application of these materials requires efforts from multiple aspects, such as improving structure order in large-scale fabrication, as well as designing compositions and tailoring properties to meet diverse requirements. Nonetheless, these materials undoubtedly offer a wide range of applications.

List of changes:

1. Slight modifications to the wording in the first paragraph of the ‘Optical transitions in *n*-stretching’ section in the main text.
2. New results have been added to the Supplementary Information (SI).

REVIEWERS' COMMENTS

Reviewer #2 (Remarks to the Author):

The authors have carefully addressed my concerns and I am satisfied with the revision.

Therefore, I would like to recommend the publication of this work in Nature Communications without further revision.

Responses to reviewers' comments:

Reviewer #2:

Comments:

The authors have carefully addressed my concerns and I am satisfied with the revision. Therefore, I would like to recommend the publication of this work in Nature Communications without further revision.

R: We greatly appreciate the reviewer's comments. We would like to thank the reviewer for the valuable feedback during the review process